# Neural Stochastic PDEs: Resolution-Invariant Learning of Continuous Spatiotemporal Dynamics

**Cristopher Salvi**
Imperial College London &
The Alan Turing Institute
c.salvi@imperial.ac.uk

**Maud Lemercier**
University of Warwick
maud.lemercier@warwick.ac.uk

**Andris Gerasimovičs**
University of Bath
ag2616@bath.ac.uk

## Abstract

*Stochastic partial differential equations* (SPDEs) are the mathematical tool of choice for modelling spatiotemporal PDE-dynamics under the influence of randomness. Based on the notion of mild solution of an SPDE, we introduce a novel neural architecture to learn solution operators of PDEs with (possibly stochastic) forcing from partially observed data. The proposed *Neural SPDE* model provides an extension to two popular classes of physics-inspired architectures. On the one hand, it extends Neural CDEs and variants – continuous-time analogues of RNNs – in that it is capable of processing incoming sequential information arriving at arbitrary spatial resolutions. On the other hand, it extends Neural Operators – generalizations of neural networks to model mappings between spaces of functions – in that it can parameterize solution operators of SPDEs depending simultaneously on the initial condition and a realization of the driving noise. By performing operations in the spectral domain, we show how a Neural SPDE can be evaluated in two ways, either by calling an ODE solver (emulating a spectral Galerkin scheme), or by solving a fixed point problem. Experiments on various semilinear SPDEs, including the stochastic Navier-Stokes equations, demonstrate how the Neural SPDE model is capable of learning complex spatiotemporal dynamics in a resolution-invariant way, with better accuracy and lighter training data requirements compared to alternative models, and up to 3 orders of magnitude faster than traditional solvers.

## 1 Introduction

*Stochastic partial differential equations* (SPDEs) are the mathematical formalism used to model many physical, biological and economic systems subject to the influence of randomness, be it intrinsic (e.g. quantifying uncertainty) or extrinsic (e.g. modelling environmental random perturbations). Notable examples of SPDEs include the *Kardar–Parisi–Zhang (KPZ) equation* for modelling random interface growth such as the propagation of a forest fire from a burnt region to an unburnt region [12], the *Ginzburg-Landau equation* describing phase transitions of ferromagnets and superconductors near critical temperature [34], or the *stochastic Navier-Stokes equations* modelling the dynamics of a turbulent fluid flow under the presence of local random fluctuations [31]. For an introduction to the theory of SPDEs see Hairer [11]; a comprehensive textbook is Holden et al. [15].

Classical numerical approaches for solving SPDEs include finite difference methods and spectral Galerkin methods [27] among others. To ensure accuracy and stability of numerical solutions of complex SPDEs, computations must be carried out at high resolution using fine discretization grids, rendering the resulting schemes computationally intractable. This limitation motivates the study of data-driven methods that can learn solutions to differential equations from partially observed data.

36th Conference on Neural Information Processing Systems (NeurIPS 2022).

**Related work**  There has been an increased interest in recent years to combine neural networks and differential equations into a hybrid approach [5, 17, 37].

*Neural controlled differential equations* (Neural CDEs), as popularised by [3, 18, 32], are continuous-time analogues to recurrent neural networks (RNN, GRU, LSTM etc.). The input to a Neural CDE model is a multivariate time series interpolated into a continuous path $X : [0, T] \to \mathbb{R}^{d_\xi}$; the model consists of a matrix-valued feedforward neural network $f_\theta : \mathbb{R}^{d_h} \to \mathbb{R}^{d_h \times d_\xi}$ parameterizing the vector field of the following dynamical system (and satisfying some minimal Lipschitz regularity to ensure existence and uniqueness of solutions)

$$z_0 = \ell_\theta(u_0), \quad z_t = z_0 + \int_0^t f_\theta(z_s)dX_s, \quad u_t = \pi_\theta(z_t), \tag{1}$$

where $\ell_\theta : \mathbb{R}^{d_u} \to \mathbb{R}^{d_h}$ and $\pi_\theta : \mathbb{R}^{d_h} \to \mathbb{R}^{d_u}$ are feedforward neural networks. The output response $u : [0, T] \to \mathbb{R}^{d_u}$ is then fed to a (possibly pathwise) loss function (mean squared, cross entropy etc.) and trained via stochastic gradient descent in the usual way. In practice, the term "$dX_t$" means that the solution $z_t$ of the equation (e.g. attention required from a doctor) can change in response to a change of an external stream of information $X_t$ (e.g. heart rate of a patient).

Depending on the level of roughness of the control path $X$, the integral in eq. (1) can be interpreted in different ways. In [18], $X$ is assumed differentiable and is obtained in practice via cubic splines interpolation of the original time series. In this way, the term "$dX_s$" can be interpreted as "$\dot{X}_s ds$" so that eq. (1) becomes an ODE of the form $\dot{z}_t = f_\theta(z_t)\dot{X}_t$ that can be evaluated numerically via a call to an ODE solver of choice (Euler, Runge-Kutta, implicit, adaptive stepsize schemes etc.). More generally, if $X$ is of bounded variation then the integral above can be seen as a classical Riemann–Stieltjes or Young integral [38]. Neural SDEs [19, 20, 22, 26] are a special subclass of Neural CDEs where the control is a sample path from a $d_\xi$-dimensional Brownian motion (which is not of bounded variation), and eq. (1) is understood via stochastic integration (Itô, Stratonovich etc.). Neural RDEs [32] allow to relax even further the regularity assumptions on $X$ by treating the integral using rough integration [10, 30]. In practice, Neural RDEs are particularly well suited for long time series. This is due to the fact that the model can be evaluated via a numerical scheme from stochastic analysis (called the *log-ODE method* [32]) over intervals much larger than what would be expected given the sampling rate of the time series. However, the space complexity of the numerical solver increases exponentially in the number of channels $d_\xi$, thus model complexity becomes intractable for high dimensional time series. Despite offering many advantages for modelling temporal dynamics, these models are not designed to process signals varying both in space and in time such as physical fields described by SPDEs. In particular, although these models are time-resolution invariant, they are not space-resolution invariant, and are not well suited to capture nonlinear interactions between the various space-time points typically observed in SPDE-dynamics. Similar to CDEs, the solution $u$ to an SPDE is characterized by an initial condition $u_0$ and a driving noise $X$. However, in the case of CDEs, $(u_0, X_t, u_t)$ are vectors, while in the case of SPDEs they are functions.

*Neural Operators* [21, 23, 24, 28] are generalizations of neural networks capable of modelling mappings between spaces of functions and offer an attractive option for learning with spatiotemporal data [21]. Among all kinds of Neural Operators, *Fourier Neural Operators* (FNOs) [25] stand out because of their easier parametrization while demonstrating similar learning performance compared to other Neural Operator models. However, Neural Operators generally fail to incorporate the effect that an external (possibly random) spatiotemporal signal might have on the system they describe. In the case of SPDEs, the external signal is indeed random (e.g. sample from a Wiener process) and its presence leads to new phenomena, both at the mathematical and the physical level, often describing more complex and realistic dynamics than the ones arising from deterministic PDEs.

**Contributions**  To overcome the above limitations faced by Neural CDEs and Neural Operators, we introduce the *neural stochastic partial differential equation* (Neural SPDE) model, capable of learning solution operators of SPDEs from partially observed data by processing, continuously in time and space, incoming sequential information arriving at an arbitrary resolution. We propose two separate algorithms to evaluate our model: the first reduces the Neural SPDE to a system of ODEs in Fourier space, which can then be solved numerically by means of any ODE solver of choice (emulating a spectral Galerkin scheme); the second rewrites the Neural SPDE as a fixed point problem, which is solved via classical root-finding schemes. For both choices of evaluation, the Neural SPDE model inherits memory-efficient backpropagation capabilities provided by existing

adjoint-based and implicit-differentiation-based methods respectively. Finally, we perform extensive experiments on various semilinear SPDEs, including the stochastic Ginzburg-Landau, Korteweg-De Vries, Navier-Stokes equations. The empirical results illustrate several useful aspects of our model: 1) it is space and time resolution-invariant, meaning that even if trained on a lower resolution it can be directly evaluated on a higher resolution; 2) it requires a lower amount of training data to achieve similar or better performance compared to alternative models; 3) its evaluation is up to 3 orders of magnitude faster than traditional numerical solvers.

The outline of the paper is as follows: in Sec. 2 we provide a brief introduction to SPDEs which will help us to define our Neural SPDE model in Sec. 3, followed by numerical experiments in Sec. 4. In Appendix A, we provide an overview of the computational aspects of SPDEs used to design the model and solve SPDEs numerically. Additional experiments can be found in Appendix B.

## 2 Background on SPDEs

Let $T > 0$ and $d, d_u, d_\xi \in \mathbb{N}$. Let $\mathcal{D} \subset \mathbb{R}^d$ be a bounded domain. Let $\mathcal{H}_u = \{f : \mathcal{D} \to \mathbb{R}^{d_u}\}$ and $\mathcal{H}_\xi = \{f : \mathcal{D} \to \mathbb{R}^{d_\xi}\}$ be two Hilbert spaces of functions from $\mathcal{D}$ to $\mathbb{R}^{d_u}$ and $\mathbb{R}^{d_\xi}$ respectively. We consider a large class of SPDEs of the following type

$$du_t = (\mathcal{L}u_t + F(u_t))\, dt + G(u_t)dW_t, \tag{2}$$

where $W_t$ is either an infinite dimensional $Q$-Wiener process [27, Def. 10.6] or a cylindrical Wiener process [11, Def. 3.54] with values in $\mathcal{H}_\xi$, $F : \mathcal{H}_u \to \mathcal{H}_u$ and $G : \mathcal{H}_u \to L(\mathcal{H}_\xi, \mathcal{H}_u)$ are two continuous operators, $L(\mathcal{H}_\xi, \mathcal{H}_u)$ is the space of bounded linear operators from $\mathcal{H}_\xi$ to $\mathcal{H}_u$, and $\mathcal{L}$ is a linear differential operator generating a *semigroup*[1] $e^{t\mathcal{L}} : \mathcal{H}_u \to \mathcal{H}_u$. For further details on Wiener processes see Appendix A.2, and for a primer on semigroup theory see Hairer [11, Section 4]. A function $u : [0, T] \to \mathcal{H}_u$ is said to be a *mild solution* of the SPDE (2) if for any $t \in [0, T]$ it satisfies

$$u_t = e^{t\mathcal{L}}u_0 + \int_0^t e^{(t-s)\mathcal{L}}F(u_s)ds + \int_0^t e^{(t-s)\mathcal{L}}G(u_s)dW_s,$$

where the second integral is a stochastic integral interpreted in the Itô sense [11, Def. 3.57]. Thus, an SPDE can be informally thought of as an SDE with values in the functional space $\mathcal{H}_u$ and driven by an infinite dimensional Brownian motion $W$. Assuming global Lipschitz regularity on $F$ and $G$, a mild solution $u$ to (2) exists and is unique [11, Thm. 6.4], at least for short times.

We follow Friz and Hairer [9] and consider a regularization $W^\epsilon = \varphi^\epsilon * W$ of the driving noise $W$ with a mollifier[2], where $*$ means convolution. As done in Kidger et al. [18] for Neural CDEs, we can rewrite the mild solution of the mollified version of eq. (2) as the following randomly forced PDE

$$u_t = e^{t\mathcal{L}}u_0 + \int_0^t e^{(t-s)\mathcal{L}}H_\xi(u_s)ds, \quad H_\xi(u_t) := F(u_t) + G(u_t)\xi_t, \tag{3}$$

where $\xi = \dot{W}^\epsilon$, $\mathcal{H}_u = L^2(\mathcal{D}, \mathbb{R}^{d_u})$ and $\mathcal{H}_\xi = L^2(\mathcal{D}, \mathbb{R}^{d_\xi})$. We will refer to $\xi$ as *white noise* if $W$ is a cylindrical Wiener process and as *coloured noise* if $W$ is a $Q$-Wiener process.

In view of machine learning applications, one should think of $W$ as a continuous space-time embedding of an underlying spatiotemporal data stream. In this paper we are only going to consider $W$ to be a sample path from a Wiener process, but we emphasise that the Neural SPDE model extends, in principle, beyond the scope of SPDEs and could be used for example to process videos in computer vision applications, which we leave as future work. Next we introduce the Neural SPDE model.

---

[1] A strongly continuous semigroup $S$ on $\mathcal{H}_u$ is a family of bounded linear operators $S = \{S_t : \mathcal{H}_u \to \mathcal{H}_u\}_{t \geq 0}$ with the properties that: 1) $S_0 = \text{Id}$, the identity operator on $\mathcal{H}_u$, 2) $S_t \circ S_s = S_{t+s}$, for any $s, t \geq 0$, and 3) the function $t \mapsto S_t u$ is continuous from $[0, T]$ to $\mathcal{H}_u$, for any $u \in \mathcal{H}_u$.

[2] A mollifier $\varphi$ is a smooth function on $\mathbb{R}^{d+1}$ that is: 1) compactly supported, 2) $\int_{\mathbb{R}^{d+1}} \varphi(x)dx = 1$, and 3) $\lim_{\epsilon \to 0} \varphi^\epsilon(x) = \lim_{\epsilon \to 0} \epsilon^{-(d+1)}\varphi(x/\epsilon) = \delta(x)$, where $\delta$ is the Diract delta function and the limit must be understood in the space of Schwartz distributions.

Figure 1: A signal $\xi$ and an initial condition $u_0$ are observed on a, possibly irregular, spatiotemporal grid. $u_0$ is lifted to $z_0$ living in a latent space. The solution of an SPDE in latent space driven by the signal $\xi$ and with initial condition $z_0$ is obtained using either a fixed point solver or an ODE solver. The solution $z$ is then projected back to the physical space by a linear readout. The output signal $u$ is then fed to a pathwise loss function.

## 3  Neural SPDEs

For a large class of differential operators $\mathcal{L}$, the action of the semigroup $e^{t\mathcal{L}}$ can be written as an integral against a kernel function $\mathcal{K}_t : \mathcal{D} \times \mathcal{D} \to \mathbb{R}^{d_u \times d_u}$ such that

$$(e^{t\mathcal{L}}h)(x) = \int_{\mathcal{D}} \mathcal{K}_t(x,y)h(y)\mu_t(dy),$$

for any $h \in \mathcal{H}_u$, any $x \in \mathcal{D}$ and $t \in [0,T]$, and where $\mu_t$ is a Borel measure on $\mathcal{D}$. As in Kovachki et al. [21], here we take $\mu_t$ to be the Lebesgue measure on $\mathbb{R}^d$ but other choices can be made, for example to incorporate prior information. We assume that $\mathcal{K}$ is stationary so that eq. (3) can be rewritten in terms of the spatial convolution $*$

$$u_t = \mathcal{K}_t * u_0 + \int_0^t \mathcal{K}_{t-s} * H_\xi(u_s)ds.$$

For a large class of SPDEs of the form (2), both $F$ and $G$ are local operators acting on a function $h \in \mathcal{H}_u$. In other words, the evaluations $F(h)(x)$ and $G(h)(x)$ at any point $x \in \mathcal{D}$ only depend $h(x)$, and not on the evaluation $h(y)$ at some other point $y \in \mathcal{D}$ in the neighbourhood of $x$.

### 3.1  The model

Let $\mathcal{H}_h = L^2(\mathcal{D}, \mathbb{R}^{d_h})$ for some latent space dimension $d_h > d_u$. Let

$$L_\theta : \mathbb{R}^{d_u} \to \mathbb{R}^{d_h}, \quad F_\theta : \mathbb{R}^{d_h} \to \mathbb{R}^{d_h}, \quad G_\theta : \mathbb{R}^{d_h} \to \mathbb{R}^{d_h \times d_\xi}, \quad \Pi_\theta : \mathbb{R}^{d_h} \to \mathbb{R}^{d_u}$$

be four feedforward neural networks. For any differentiable control $\xi : [0,T] \to \mathcal{H}_\xi$, define the map $H_{\theta,\xi} : \mathcal{H}_u \to \mathcal{H}_u$ so that for any $h \in \mathcal{H}_u$, $x \in \mathcal{D}$, $t \in [0,T]$

$$H_{\theta,\xi}(h)(x) = F_\theta(h(x)) + G_\theta(h(x))\xi_t,$$

A Neural SPDE is defined as follows

$$z_0(x) = L_\theta(u_0(x)), \quad z_t = \mathcal{K}_t * z_0 + \int_0^t \mathcal{K}_{t-s} * H_{\theta,\xi}(z_s)ds, \quad u_t(x) = \Pi_\theta(z_t(x)). \tag{4}$$

We note that globally Lipschitz conditions can be imposed by using ReLU or tanh activation functions in the neural networks $F_\theta$ and $G_\theta$. In secs. 3.2 and 3.3 we propose two distinct algorithms to evaluate the Neural SPDE model (4) which are based on two different parameterization of the kernel $\mathcal{K}$.

## 3.2 Evaluating the model by solving a system of ODEs

Levereraging the *convolution theorem*, we can rewrite the integral in eq. (4) as follows

$$z_t = \mathcal{F}^{-1}\Big(\mathcal{F}(\mathcal{K}_t)\mathcal{F}(z_0) + \int_0^t \mathcal{F}(\mathcal{K}_{t-s})\mathcal{F}(H_{\theta,\xi}(z_s))ds\Big),$$

where $\mathcal{F}, \mathcal{F}^{-1}$ are the $d$-*dimensional Fourier transform* (FT) and its inverse (see Def. A.1). If one further assumes that $\mathcal{L}$ is a polynomial differential operator, it can be shown that there exists a map $A: \mathbb{C}^d \to \mathbb{C}^{d_h \times d_h}$ such that $\mathcal{F}(\mathcal{K}_t)(y) = e^{tA(y)}$ (see appendix B.1 for a derivation). It follows that

$$z_t = \mathcal{F}^{-1}\Big(e^{tA}\mathcal{F}(z_0) + \int_0^t e^{(t-s)A}\mathcal{F}(H_{\theta,\xi}(z_s))ds\Big) = \mathcal{F}^{-1}(v_t),$$

where $v_t: \mathbb{C}^d \to \mathbb{C}^{d_h}$ is the solution of the following ODE

$$v_t = v_0 + \int_0^t Av_s + \mathcal{F}(H_{\theta,\xi}(\mathcal{F}^{-1}(v_s))).$$

Hence, $z_t$ can be obtained by applying the inverse FT to the output of an ODE solver on $[0, t]$ with initial condition $\mathcal{F}(z_0)$, vector field $\Psi_{\theta,\xi} := A + \mathcal{F} \circ H_{\theta,\xi} \circ \mathcal{F}^{-1}$, i.e.

$$z_t \approx \mathcal{F}^{-1}\big(\text{ODESolve}(\mathcal{F}(z_0), \Psi_{\theta,\xi}, [0, t])\big)$$

This approach can naturally be seen as a "neural version" of the classical *spectral Galerkin method* for SPDEs as described in Appendix A.3.

We note that this numerical evaluation of the Neural SPDE model (4) allows to inherit memory-efficient adjoint-based backpropagation capabilities as in the evaluation of a Neural CDE. For further details on adjoint-based backpropagation we refer the reader to Chen et al. [5], Kidger [17].

## 3.3 Evaluating the model by solving a fixed point problem

In our second approach of model evaluation, we make use of three different versions of the FT: the time-only FT $\mathcal{F}_1$ and its inverse $\mathcal{F}_1^{-1}$, the space-only FT $\mathcal{F}_d$ and its inverse $\mathcal{F}_d^{-1}$, and the space-time FT $\mathcal{F}_{d+1}$ and its inverse $\mathcal{F}_{d+1}^{-1}$ (see Def. A.1 for details). Denoting by $\star$ the space-time convolution, the integral in eq. (4) can be rewritten as

$$z_t = \mathcal{K}_t * z_0 + (\mathcal{K} \star \mathbb{1}_{\geq 0}H_{\theta,\xi}(z.))_t,$$

where $\mathbb{1}_{\geq 0}$ is the indicator function restricting the temporal domain to the positive real line. Using again the convolution theorem we obtain

$$z_t = \mathcal{F}_d^{-1}(\mathcal{F}_d(\mathcal{K}_t)\mathcal{F}_d(z_0)) + \mathcal{F}_{d+1}^{-1}(\mathcal{F}_{d+1}(\mathcal{K})\mathcal{F}_{d+1}(\mathbb{1}_{\geq 0}H_{\theta,\xi}(z.)))_t,$$

where all multiplications are matrix-vector multiplications. Using the trick introduced in [25], one can parameterize $\mathcal{F}_{d+1}(\mathcal{K})(y)$ directly in Fourier space as a complex tensor $B$, so that the solution of eq. (4) can be obtained by solving the fixed point problem $z = \Phi_{\theta,\xi}(z)$ with

$$\Phi_{\theta,\xi}(z)_t := \mathcal{F}_d^{-1}\big(\mathcal{F}_1^{-1}(B)_t\mathcal{F}_d(z_0)\big) + \mathcal{F}_{d+1}^{-1}\big(B\mathcal{F}_{d+1}(\mathbb{1}_{\geq 0}H_{\theta,\xi}(z.))\big)_t,$$

and where we used the fact that $\mathcal{F}_d(\mathcal{K}_t) = \mathcal{F}_1^{-1}(\mathcal{F}_{d+1}(\mathcal{K}))_t$. This can be solved numerically using classical root-finding schemes (e.g. by Picard's iteration)

$$z \approx \text{FixedPointSolve}(z_0, \Phi_{\theta,\xi}).$$

Analogously to adjoint-based backpropagation for the evaluation approach mentioned in Sec. 3.2, there is a mechanism that leverages the *implicit function theorem* allowing to backpropagate through the operations of a fixed point solver in a memory-efficient way. See Bai et al. [2] for further details.

We note that the FTs are numerically approximated using the *discrete Fourier transform* (DFT) and selecting a maximum number of frequency modes [3] (see Appendix A.1 for details).

---

[3] The DFT approximates the Fourier series expansion truncated at a maximum number of modes. This allows to specify the shape of the two complex tensors $A$ ($k_{\max}^1 \times \ldots \times k_{\max}^d \times d_h \times d_h$) in Sec. 3.2 and $B$ ($k_{\max}^1 \times \ldots \times k_{\max}^{d+1} \times d_h \times d_h$) in Sec. 3.3. The $k_{\max}^i$ are treated as hyperparameters of the model.

### 3.4 Space-time resolution-invariance

As depicted in Fig. 1, the input to a Neural SPDE corresponds to a (possibly irregularly sampled) time-indexed sequence of (possibly partially-observed) spatial observations recorded on a space-time grid. The data is then interpolated into a continuous spatiotemporal signal $\xi$ and initial condition $u_0$. By construction, a Neural SPDE operates in continuous time and space on the tuple of functions $(u_0, \xi)$ and produces a spatiotemporal response $u$, which is also continuous in space and time; the function $u$ can then be evaluated at an arbitrary space-time resolution, possibly different from the one used during training. In the next section we will demonstrate empirically that even if trained on a coarser resolution, a Neural SPDE can be evaluated on a finer resolution without sacrificing performance, a property known as *zero-shot super-resolution*.

### 3.5 Comparison of the two evaluation methods

**Number of parameters**    For a fixed dimension $d_h$ of the latent space, the majority of trainable parameters in the ODE parameterization lies in the complex tensor $A$, which consists of $k_{\max}^1 ... k_{\max}^d d_h^2$ parameters, where each $k_{\max}^i$ is the maximum number of selected frequencies in the Fourier domain. Regarding the Fixed Point parameterization, the bulk of the parameters is in the complex tensor $B$, which consists of $k_{\max}^1 ... k_{\max}^{d+1} d_h^2$, where the additional frequency is due to the fact that we are taking the FFT in space-time rather than just in space as done in the ODE Solver approach. Hence, the latter would in principle have the advantage of using a lower number of parameters than the former; however, to achieve similar performance, we found that the dimensionality of the latent space has to be roughly 20 times higher, which offsets the aforementioned advantage.

**Time complexities**    The time complexity of the ODE Solver approach is $\mathcal{O}(NN_x \log(N_x))$, where $N$ is the number of time steps taken by the ODE solver and $N_x$ is the number of points on the spatial grid, while the complexity of the Fixed Point approach is $\mathcal{O}(IN_xN_t(\log(N_x) + \log(N_t)))$ where $I$ is the number of Picard iterations and $N_t$ is the number of points on the temporal grid. In our experiments we choose $N_t \approx N_x$ and $IN_t \approx N$, making the two complexities comparable.

**Speed of computation**    We found that the ODE approach is approximately 10 times slower than the Fixed Point approach. We believe this is largely an implementation issue of the `torchdiffeq` library, while the FFT is a highly optimised transform in Pytorch.

### 3.6 Considerations about convergence

We follow Friz and Hairer [9] and consider a regularization $W^\epsilon = \varphi^\epsilon * W$ of the driving noise $W$ with a compactly supported smooth mollifier $\varphi^\epsilon$. It is a classical result (Wong-Zakai [35]) from rough path theory [10, 30] that, for the case of SDEs, the sequence of random ODEs driven by the mollification of Brownian motion converges in probability to a limiting process that does not depend on the choice of mollifier and agrees with the Stratonovich solution of the SDE. Furthermore, the solution map $(u_0, W) \mapsto u$ is continuous in an appropriate rough path topology. This result nicely extends to the setting of SPDEs driven by a finite dimensional noise [9, Thm. 1.3]: if $u^\epsilon$ denotes the random PDE solutions driven by $\dot{W}^\epsilon dt$ (instead of $\circ dW_t$), then $u^\epsilon$ converges in probability to a limiting process corresponding to the Stratonovich solution of the SPDE. In our setting though, the driving noise is infinite dimensional and the resulting integral cannot be interpreted in the Stratonovich sense because otherwise the corresponding Itô-Stratonovich correction would be infinite. Nonetheless, Hairer and Pardoux [14, Thm. 1.1] show that, in the case of the heat operator and under appropriate renormalization and drift correction, the random PDE solution $u^\epsilon$ converges in probability to the Itô solution of the SPDE, and that the solution map is continuous in an appropriate regularity structures topology. We note that extending this result to a generic differential operators would require a similarly rigorous proof, which goes beyond the scope of this article and that we leave as future work.

## 4 Experiments

In this section, we run experiments on three semilinear SPDEs: the stochastic Ginzburg-Landau equation in 4.1, the stochastic Korteweg-De Vries equation in 4.2, and the stochastic Navier-Stokes equations in 4.3. We note that although the assumption of globally Lipschitz vector fields might be

violated for the following SPDEs, well-posedness (i.e. existence of global solutions) can be shown using equation-specific arguments. We consider three supervised operator-learning settings:

- $u_0 \mapsto u$, assuming the noise $\xi$ is not observed;
- $\xi \mapsto u$, assuming the noise $\xi$ is observed, but the initial condition $u_0$ is fixed across samples;
- $(u_0, \xi) \mapsto u$, assuming the noise $\xi$ is observed and $u_0$ changes across samples.

We note that learning the operator $u_0 \mapsto u$ of an SPDE without observing the driving noise $\xi$ unavoidably yields poor results for all considered models as only partial information about the system is provided as input. However, we find it informative to include the performances obtained in this setting, as this provides a sanity check that emphasizes the importance of the noise in all the experiments we consider in this paper. Moreover, the ability to process the initial condition $u_0$ on its own (in absence of noise) testifies that Neural SPDEs can also be used to learn deterministic PDEs. We provide an example on the deterministic Navier-Stokes equations in Appendix B.5.

Neural CDE, Neural RDE, FNO and DeepONet [28, 29] will be the main benchmark models. In addition, we also propose an additional baseline Neural CDE-FNO, which is a hybrid model consisting of a Neural CDE where the drift is modelled by an FNO and the diffusion by a feedforward neural network. The motivation for using a FNO to represent the drift comes from the universal approximation properties of FNOs studied in Kovachki et al. [21, Thm. 4].

An interesting line of work to tackle SPDE-learning is provided in [7, 16]. The authors construct a set of features from the pair $(u_0, \xi)$ following the definition of a model from the theory of regularity structures [13]. They then perform linear [7] and nonlinear [16] regression from these features to the solution of the SPDE at a single time point. Therefore, these models would have to be retrained for any new prediction. In addition, both [7, 16] assume knowledge of the differential operator $\mathcal{L}$ governing the dynamics, while Neural SPDE learns a representation of $\mathcal{L}$ via the parametrization of the associated kernel. For these reasons these recent models are not included in our benchmark.

For all the experiments, the loss function is the relative pathwise $L^2$ error. The hyper-parameters for all the models are selected by grid-search (see Appendix B.2 for further experimental details). Experiments are run on a Tesla P100 NVIDIA GPU. The code for the experiments is provided in the supplementary material. Additional experiments may be found in Appendix B.

### 4.1 Stochastic Ginzburg-Landau equation

We start with the stochastic Ginzburg-Landau equation, a reaction diffusion equation in 1D given by

$$\partial_t u - \Delta u = 3u - u^3 + \xi, \qquad u(t, 0) = u(t, 1), \quad u(0, x) = u_0(x), \quad (t, x) \in [0, T] \times [0, 1].$$

This equation is also known as the Allen-Cahn equation in 1-dimension and is used for modeling various physical phenomena like superconductivity [34]. Here $\xi$ denotes space-time white noise with sample paths generated using classical sampling schemes for Wiener processes detailed in A.2.

Table 1: **Ginzburg-Landau**. Relative L2 error on the test set. x indicates that the model is not applicable.

| Model | $N = 1\,000$ | | | $N = 10\,000$ | | |
|---|---|---|---|---|---|---|
| | $u_0 \mapsto u$ | $\xi \mapsto u$ | $(u_0, \xi) \mapsto u$ | $u_0 \mapsto u$ | $\xi \mapsto u$ | $(u_0, \xi) \mapsto u$ |
| NCDE | x | 0.112 | 0.127 | x | 0.056 | 0.072 |
| NRDE | x | 0.129 | 0.150 | x | 0.070 | 0.083 |
| NCDE-FNO | x | 0.071 | 0.066 | x | 0.066 | 0.069 |
| DeepONet | 0.130 | 0.126 | x | 0.126 | 0.061 | x |
| FNO | 0.128 | 0.032 | x | 0.126 | 0.027 | x |
| NSPDE (Ours) | 0.128 | **0.009** | **0.012** | 0.126 | **0.006** | **0.006** |

We consider two data-regimes: a *low data regime* where the total number of training observations is $N = 1\,000$, and a *large data regime* where $N = 10\,000$. In both cases, the response paths are generated by solving the SPDE along each sample path of the noise $\xi$ using a finite difference

scheme described in Appendix A.3 using 128 evenly distanced points in space and time and step size $\Delta t = 10^{-3}$. Following the same setup as in Chevyrev et al. [7, eq. (3.6)], we solve the SPDE until $T = 0.05$ resulting in 50 time points . We choose as initial condition $u_0(x) = x(1-x) + \kappa\eta(x)$, with $\eta(x) = a_0 + \sum_{k=-10}^{k=10} a_k/(1 + |k|^2) \sin(k\pi x)$ where $a_k \sim \mathcal{N}(0,1)$. We take $\kappa = 0$ and $\kappa = 0.1$ to generate a dataset where the initial data is either fixed or varies across samples. We provide extra experiments on this SPDE for larger time horizons $T$ and multiplicative forcing in Appendix B.3. We report the results in Table 1. The Neural SPDE model (NSPDE) yields the lowest relative error for all tasks, reaching one order of magnitude improvement on the main task $(u_0, \xi) \mapsto u$ in the large data regime compared to all the applicable benchmark models (NCDE, NRDE, NCDE-FNO). In all settings, even with a limited amount of training samples ($N = 1\,000$), NSPDE achieves $\sim 1\%$ error rate, and marginally improves to $< 1\%$ error when $N = 10\,000$.

## 4.2 Stochastic Korteweg–De Vries equation

Next, we consider the stochastic Korteweg–De Vries (KdV) equation, a higher order SPDE given by

$$\partial_t u + \gamma\partial_x^3 u = 6u\partial_x u + \xi, \qquad u(t,0) = u(t,1), \quad u(0,x) = u_0(x), \quad (t,x) \in [0,T] \times [0,1].$$

This equation is used to describe the propagation of nonlinear waves at the surface of a fluid subject to random perturbations (another wave equation is studied in Appendix B.4). We refer the reader to Wazwaz [36] for an overview on the KdV equation and its relations to solitary waves. The stochastic forcing is given by $\xi = \dot{W}$ for $W$ being a partial sum approximation of a Q-Wiener process as per Example 10.8 in Lord et al. [27] with $\lambda_j \sim j^{-5+\varepsilon}$ and $\phi_j(x) = \sin(j\pi x)$ (see eq. (6) in Appendix A.2). Taking small $\varepsilon > 0$ guarantees that $W_t$ is twice differentiable in space for every $t \geq 0$. To generate the datasets, we solve the SPDE with $\gamma = 0.1$ until $T = 0.5$.

Table 2: **Stochastic KdV**. Relative L2 error on the test set. The symbol x indicates that the model is not applicable.

(a) $N = 1\,000$ and $T = 0.5$.

| Model | $u_0 \mapsto u$ | $\xi \mapsto u$ | $(u_0, \xi) \mapsto u$ |
|---|---|---|---|
| NCDE | x | 0.464 | 0.466 |
| NRDE | x | 0.497 | 0.503 |
| NCDE-FNO | x | 0.126 | 0.259 |
| DeepONet | 0.874 | 0.235 | x |
| FNO | 0.835 | 0.079 | x |
| NSPDE (Ours) | 0.832 | **0.004** | **0.008** |

(b) $N = 1\,000$ and $T = 1$.

| Model | $u_0 \mapsto u$ | $\xi \mapsto u$ | $(u_0, \xi) \mapsto u$ |
|---|---|---|---|
| FNO | 0.913 | 0.112 | x |
| NSPDE (Ours) | 0.904 | **0.009** | 0.012 |

(c) Subsampling ($N = 1\,000$ and $T = 0.5$).

| Subsampling rates | | $\xi \mapsto u$ | $(u_0, \xi) \mapsto u$ |
|---|---|---|---|
| Time : 0% | Space : 0% | 0.004 | 0.008 |
| Time : 10% | Space : 0% | 0.076 | 0.059 |
| Time : 0% | Space : 50% | 0.005 | 0.008 |

The stochastic forcing is simulated using 128 evenly distanced points in space and a time step $\Delta t_{\text{ref}} = 10^{-3}$. We then approximate realizations of the solution of the KdV equation using a time step $\Delta t = 10^{-2}$ until $T = 0.5$. Here, the initial condition is given by $u_0(x) = \sin(2\pi x) + \kappa\eta(x)$, where $\eta$ is defined as in Sec. 4.1. Similarly to Sec. 4.1 we either take $\kappa = 0$ or $\kappa = 1$ to generate datasets where the initial condition is either fixed or varies across samples. Each dataset consists of $N = 1\,000$ training observations. As reported in Table 2a, Neural SPDEs outperforms the second best model FNO by a full order of magnitude in the task $\xi \mapsto u$ and the second best model NCDE-FNO by almost two orders of magnitude in the task $(u_0, \xi) \mapsto u$. We also perform the same tasks for a larger time horizon $T = 1$ and report the results of a comparison against FNO in Table 2b.

**Partial observations** Neural SPDEs are able to process signals that are irregularly sampled both in space and in time by interpolating between observations. Yet, the ability of a model to process irregular data does not guarantee its robustness when some observations are dropped. Robustness can only be guaranteed if the signal is regular enough so that replacing dropped observations by interpolation results in a new signal that is close, in some suitable norm, to the original signal. To illustrate this point, we run two additional experiments where we drop uniformly at random 1) 10% of the data in time and 2) 50% of the data in space. As it can be observed in Table 2c, the performance of Neural SPDE remains roughly unchanged when data is dropped in space but decreases when data

is dropped in time, which is to be expected since the driving signal is a Q-Wiener process, which is rough in time, but smoother in space. We also note that to ensure a good approximation of the FT by the FFT, the interpolation must translate the irregular data to a (possibly finer) regular grid.

### 4.3 Stochastic Navier-Stokes equations in 2D

Finally, we consider the vorticity form of the Navier-Stokes equations for an incompressible flow

$$\partial_t w - \nu \Delta w = -u \cdot \nabla w + f + \sigma \xi, \qquad w(0, x) = w_0(x), \quad (t, x) \in [0, T] \times [0, 1]^2, \quad (5)$$

where $u$ is the unique divergence free ($\nabla \cdot u = 0$) velocity field such that $w = \nabla \times u$. These equations describe the motion of an incompressible fluid with viscosity $\nu$ subject to external forces [34]. The deterministic forcing $f$, defined as in Li et al. [25], is a function of space only. The stochastic forcing $\xi$ is given by $\xi = \dot{W}$ for $W$ being a Q-Wiener process which is colored in space and rescaled by $\sigma = 0.05$ (see Appendix A.2). The initial condition is generated according to $w_0 \sim \mathcal{N}(0, 3^{3/2}(-\Delta + 49I)^{-3})$ with periodic boundary conditions. The viscosity is set to $\nu = 10^{-4}$.

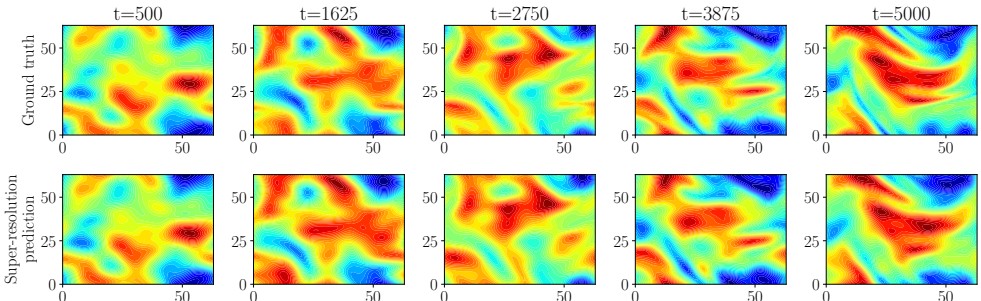

Figure 2: **Top panel:** Solution of the vorticity equation for one realisation of the stochastic forcing between the $500^{\text{th}}$ and the $5\,000^{\text{th}}$ time steps. **Bottom panel:** Predictions with the Neural SPDE model given the initial condition at the $500^{\text{th}}$ time step and the forcing between the $500^{\text{th}}$ and the $5\,000^{\text{th}}$ time steps. The model is trained on a $16 \times 16$ mesh and evaluated on a $64 \times 64$ mesh.

For each realization of the Q-Wiener process (sampled according to the scheme in Appendix A.2) we solve eq. (5) with a pseudo-spectral solver described in Appendix A.3, where time is advanced with a Crank–Nicolson update. We solve the SPDE on a $64 \times 64$ mesh in space and use a time step of size $10^{-3}$. For the tasks $u_0 \mapsto u$ and $\xi \mapsto u$, we generate the datasets by solving the SPDE up to time $T = 1$ and downsample the trajectories by a factor of 10 in time (resulting in 100 time steps) and 4 in space (resulting in a $16 \times 16$ spatial resolution). The number of training samples is $N = 1\,000$. To generate the training set for the task $(u_0, \xi) \mapsto u$, we generate 10 long trajectories of $15\,000$ steps each up to time $T = 15$. We partition each trajectory into consecutive sub-trajectories of 500 time-steps using a rolling window. This yields a total of $2\,000$ input-output pairs. We split the data into shorter sequences of 500 time steps so that one batch fits in memory of the used GPU.

Table 3: **Stochastic Navier-Stokes**. Relative L2 error on the test set. The symbol x indicates that the model is not applicable, and - indicates that the model does not fit in memory.

| Model | $u_0 \mapsto u$ | $\xi \mapsto u$ | $(u_0, \xi) \mapsto u$ |
|---|---|---|---|
| NCDE | x | 0.366 | 0.843 |
| NRDE | x | - | - |
| NCDE-FNO | x | 0.326 | 0.178 |
| DeepONet | 0.432 | 0.348 | x |
| FNO | 0.188 | 0.039 | x |
| NSPDE (Ours) | 0.155 | **0.034** | **0.049** |

As shown in Table 3, Neural SPDEs marginally outperforms FNO on the task $\xi \mapsto u$, but with a significantly larger gap on the task $(u_0, \xi) \mapsto u$ from the second best model NCDE-FNO. Fig. 2 indicates that our model is capable of zero-shot super-resolution in space-time, achieving good performance even when evaluated on a larger time horizon and on an upsampled spatial grid. Finally, we report in Table 4 some run time statistics indicating that NSPDEs can be up to 3 orders of magnitude faster than traditional numerical solvers.

## 5   Conclusion

We introduced Neural SPDEs, a model capable of learning solution operators of PDEs with (possibly stochastic) forcing from partially observed data. Our model provides an extension to two classes of physics-inspired models. It extends Neural CDEs in that it is resolution-invariant both in space and in time, and it extends Neural Operators as it can be used to learn solution operators of SPDEs depending simultaneously on the initial condition and driving noise. We performed extensive experiments illustrating how the model achieves superior performance while requiring a lower amount of training data compared to other models, and its evaluation is up to 3 orders of magnitude faster than traditional numerical solvers.

Table 4: Ratio of the inference time of the trained NSPDE over the runtime of the numerical solver. We use the same spatiotemporal discretization for both NSPDE and the numerical solver.

| Dataset | Speedup |
|---|---|
| Ginzburg-Landau | $59\times$ |
| Korteweg-De Vries | $80\times$ |
| Navier-Stokes | $300\times$ |

**Limitations and future work**   Similarly to other neural operator models, parameterising the kernel in Fourier space is by no means the only available option; other parameterisations could mitigate some of the disadvantages of the FFT (irregular grids, aliasing effect ...), see for example [23]. A question we leave to future work is how to construct a discrepancy between probability measures supported on spatiotemporal signals, generalizing for example the signature kernel MMD in [33]. Neural SPDEs paired with such discrepancy would allow the design of new generative models for spatiotemporal signals. Another research direction will be to assess whether Neural SPDEs can be used in computer vision to process videos at arbitrary resolution.

## Acknowledgments and Disclosure of Funding

This project was supported by G-Research and by DataSig under the grant EP/S026347/1.

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
