## Appendix

This appendix is organized as follows. In Appendix A we provide a summary of the computational aspects of SPDEs used for data simulation and model definition, emphasizing the important role of the Fourier Transform (A.1) for simulating noise realizations of Wiener processes (A.2) and building numerical solvers for SPDEs (A.3). In Appendix B we provide additional considerations about our Neural SPDE model and further experimental details (B.2) and additional experiments on the stochastic Ginzburg-Landau (B.3) and wave (B.4) equations, and on the deterministic Navier-Stokes PDE (B.5).

# A Computational aspects of SPDEs

We start this section with the definition of the *Fourier Transform* (FT). We then define the *Discrete Fourier Transform* (DFT) as an approximation to the FT of a function observed at finitely many locations. Next, we discuss the role played by the FT to sample realizations of Wiener processes, necessary to build spectral solvers for SPDEs. The interested reader is referred to Briggs and Henson [4] and Lord et al. [27] for further details.

## A.1 The Fourier Transform

Let $V$ be a vector space over the complex numbers (e.g. $\mathbb{C}^{d_h}$ or $\mathbb{C}^{d_h \times d_h}$). Let $r \in \mathbb{N}$ and let $\mathcal{C} \subset \mathbb{R}^r$ be a compact subset of $\mathbb{R}^r$. In the paper we used either $r = d$ and $\mathcal{C} = \mathcal{D}$ or $r = d + 1$ and $\mathcal{C} = [0, T] \times \mathcal{D}$.

**Definition A.1** (*r*-**dimensional Fourier Transform**). The $r$-dimensional FT $\mathcal{F}_r : L^2(\mathbb{R}^r, V) \rightarrow L^2(\mathbb{R}^r, V)$ and its inverse $\mathcal{F}_r^{-1} : L^2(\mathbb{R}^r, V) \rightarrow L^2(\mathbb{R}^r, V)$ are defined as follows

$$\mathcal{F}_r(f)(y) = \int_{\mathbb{R}^r} e^{-2\pi i \langle x,y \rangle} f(x) dx, \quad \mathcal{F}_r^{-1}(g)(x) = \int_{\mathbb{R}^r} e^{2\pi i \langle x,y \rangle} g(y) dy$$

for any $f, g \in L^2(\mathbb{R}^r, V)$, where $i = \sqrt{-1}$ is the imaginary unit and $\langle \cdot, \cdot \rangle$ denotes the Euclidean inner product on $\mathbb{R}^r$.

In practice, we do not observe a function on $\mathbb{R}^r$ but on a subset $\mathcal{C} \subset \mathbb{R}^r$. Furthermore, functions are observed at finitely many locations in $\mathcal{C}$, and another transform—the discrete Fourier transform (DFT)—is used for numerical computations.

In the sequel we denote by $\Pi_N$ the set of periodic sequences indexed on $\mathbb{Z}_r$ with period vector $(N_1, \ldots, N_r)$.

**Definition A.2** (*r*-**dimensional Discrete Fourier Transform**). The $r$-dimensional DFT $\mathcal{D}_r : \Pi_N \rightarrow \Pi_N$ and its inverse $\mathcal{D}_r^{-1} : \Pi_N \rightarrow \Pi_N$ are defined as follows,

$$\mathcal{D}_r(u)_n = \sum_{k \in \mathbb{Z}^r \cap \mathcal{R}_N} u_k e^{-2\pi i \langle n, N^{-1}k \rangle}, \quad \mathcal{D}_r^{-1}(v)_k = \frac{1}{|\det N|} \sum_{n \in \mathbb{Z}^r \cap \mathcal{R}_N} v_n e^{2\pi i \langle n, N^{-1}k \rangle}$$

with $N = \mathrm{diag}(N_1, \ldots, N_r) \in \mathbb{N}^{r \times r}$, and $\mathcal{R}_N$ the rectangular domain $\mathcal{R}_N = \{x \in \mathbb{R}^r \mid 0 \leq x_i < N_i, \ i = 1, \ldots, r\}$.

The DFT of a sequence can be computed exactly and efficiently using the *fast Fourier transform* (FFT) algorithm [8] which reduces the complexity from $\mathcal{O}(M^2)$ to $\mathcal{O}(M \log M)$ where $M = N_1 N_2 \ldots N_r$. Most importantly, the FFT algorithm is implemented in machine learning libraries such as PyTorch, which provide support for GPU acceleration and automatic differentiation capabilities.

Note that if we have a finite sequence, we may still define its DFT by implicitly extending the sequence periodically. In particular, when a compactly supported function is sampled on its interval of support, and the samples are used as input for a DFT, it is as if the periodic extension of the function had been sampled. More precisely, consider an input sequence which corresponds to the evaluation of a function $f$ on a regular grid of $\mathcal{C} = \mathcal{R}_N$. For simplicity, suppose that $N_i = N_1$ for all $i = 1, \ldots, r$ and consider the grid points $x_n = nL/N_1$ for $n \in \mathbb{Z}^r \cap \mathcal{R}_N$. Taking the DFT of the sequence of general term $u_n = f(x_n)$ we obtain for all $n \in \mathbb{Z}^r$,

$$\mathcal{D}_r(u)_n = \sum_{k \in \mathbb{Z}^r \cap \mathcal{R}_N} u_k e^{-2\pi i \langle n, k/N_1 \rangle} = \sum_{k \in \mathbb{Z}^r \cap \mathcal{R}_N} f(x_k) e^{-2\pi i \langle y_n, x_k \rangle},$$

where $y_n$ are the reciprocal frequency points given by $y_n = n/L$ for $n \in \mathbb{Z}^r \cap \mathcal{R}_N$. The DFT of a compactly supported (or approximately compactly supported) function $f$ sampled on the regular grid of points $x_k$ approximates the FT of $f$ at the frequency points $y_n$ (up to a constant multiplicative factor).

The FT is closely related to the notions of *Fourier coefficients* and *Fourier Series* defined hereafter.

**Definition A.3** (*r*-**dimensional Fourier series**). Let $f$ be a piecewise smooth function $f : \mathbb{R}^r \rightarrow V$ which is periodic in $x_i$ with period $L_i \in \mathbb{R}_+$ for all $i = 1, \ldots, r$. The $r$-dimensional Fourier series of

$f$ is a representation of the form,

$$f(x) \sim \sum_{n \in \mathbb{Z}^r} c_n(f) e^{2\pi i \langle L^{-1} n, x \rangle},$$

where $L = \text{diag}(L_1, \ldots, L_r) \in \mathbb{R}^{r \times r}$ and $c_n(f)$ are complex coefficients, called *Fourier coefficients*, given by

$$c_n(f) = \frac{1}{|\det L|} \int_{\mathcal{R}_L} e^{-2\pi i \langle L^{-1} n, x \rangle} f(x) dx, \quad n \in \mathbb{Z}^r$$

where $\mathcal{R}_L \subset \mathbb{R}^r$ denotes the rectangular domain of sides $L_1, \ldots, L_r$.

We note that in the definition above, the sign $\sim$ means that the series is a formal series and no statement is made about the convergence of the series (the forms of convergence are studied in Alimov et al. [1]). If $f$ is compactly supported on $\mathcal{R}_L$, we may still define its Fourier coefficients, and in this case $\mathcal{F}_r(f)(y_n) = |\det L| c_n(f)$ at the frequency points $y_n = L^{-1} n$.

**Numerical consideration**

Consider a function $f$ which has compact support (or is periodic) which is observed at $M$ locations in its support (or its unitary cell $\mathcal{R}_L$). When using the DFT to approximate $M$ points of the spectrum $\mathcal{F}_r(f)(y_k)$ (or $M$ coefficients $c_k(f)$), a so-called *aliasing* error usually occurs: due to the periodicity of the DFT, the $k^{\text{th}}$ coefficient of the DFT includes the contributions not only of the $k^{\text{th}}$ frequency mode, but also from higher modes of the underlying function $f$. In general the accuracy of the highest frequency modes is more impacted by this error, and aliasing occurs specifically when we compute nonlinear terms in the physical space. For example, in the main paper we approximate the evaluation on a discretization spatiotemporal grid $D \times \mathcal{T}$ of $\mathcal{F}_{d+1}^{-1}(\mathcal{F}_{d+1}(\mathcal{K})\mathcal{F}_{d+1}(f))$ by $\mathcal{D}_{d+1}^{-1}(B\mathcal{D}_{d+1}(f|_{D \times \mathcal{T}}))$ where $f = \mathbb{1}_{\geq 0} H_{\theta, \xi}(z)$ and $H_{\theta, \xi}$ is nonlinear. One possibility to mitigate aliasing is to set to zero the DFT terms (arising in nonlinearities) corresponding to the highest frequency modes before we apply the inverse DFT to go back to the physical space. This is precisely what we do when we parametrize only $k_{\max}^1 \times \ldots \times k_{\max}^{d+1} \times d_h \times d_h$ entries of the complex tensor $B$, and set the others to zero, hence resolving potential aliasing errors. We note that specific rules have been proposed (notably in the literature on pseudo-spectral solvers) to deal with specific nonlinearities. However, in the context of Neural SPDE we learn the nonlinearities, hence the number of frequency modes that we retain is treated as an hyperparameter.

## A.2 Stochastic simulation of Wiener processes

After defining Wiener processes we outline the sampling procedure that we used to simulate the datasets in the main paper. For more details on computational aspects of SPDEs the reader is referred to Lord et al. [27].

Throughout this section, $H$ will denote a separable Hilbert space (e.g. $H = L^2(\mathcal{D})$) with a complete orthonormal basis $\{\phi_k\}_{k \in \mathbb{N}}$. Let $(\Omega, \mathcal{F}, \mathcal{F}_t, \mathbb{P})$ be a filtered probability space.

### A.2.1 $Q$-Wiener process

Consider an operator $\mathcal{Q} : H \to H$ such that there exists a bounded sequence of nonnegative real numbers $\{\lambda_k\}_{k \in \mathbb{N}}$ such that $Q\phi_k = \lambda_k \phi_k$ for all $k \in \mathbb{N}$ (this is implied by $Q$ being a trace class, non-negative, symmetric operator, for example).

**Definition A.4** ($Q$-**Wiener process**). Let $Q$ be a trace class non negative, symmetric operator on $H$. A $H$-valued stochastic process $\{W(t) : t \geq 0\}$ is called a $Q$-Wiener process if

1. $W(0) = 0$ almost surely;

2. $W(t; \omega)$ is a continuous sample trajectory $\mathbb{R}^+ \mapsto H$, for each $\omega \in \Omega$;

3. $W(t)$ is $\mathcal{F}_t$-adapted and has independent increments $W(t) - W(s)$ for $s < t$;

4. $W(t) - W(s) \sim \mathcal{N}(0, (t - s)Q)$ for all $0 \leq s \leq t$.

In analogy to the Karhunen Loéve expansion, it can be shown that $W(t)$ is a $Q$-Wiener process if and only if for all $t \geq 0$,

$$W(t) = \sum_{j=1}^{\infty} \sqrt{\lambda_j} \phi_j \beta_j(t) \tag{6}$$

where $\beta_j(t)$ are i.i.d. Brownian motions, and the series converges in $L^2(\Omega, H)$. Moreover the series is $\mathbb{P}$-a.s. uniformly convergent on $[0, T]$ for arbitrary $T > 0$. (i.e. converges in $L^2(\Omega, \mathcal{C}([0, T], H))$).

In the Navier-Stokes example, we drive the SPDE by samples $\xi$ from a $Q$-Wiener process in two dimensions. Here we follow Lord et al. [27, Example 10.12] and explain how the sampling procedure works in this case. Let $D = (0, L_1) \times (0, L_2)$ and consider an $L^2(D)$-valued Q-Wiener process $W(t)$. If the eigenfunctions of $Q$ are given by,

$$\phi_k(x) = \frac{1}{\sqrt{L_1 L_2}} e^{2i\pi(k_1 x_1/L_1 + k_2 x_2/L_2)}$$

numerical approximation of sample paths from $W(t)$ are easy to obtain through a DFT. Denote by $\lambda_k$ the eigenvalues of $Q$ (e.g. $\lambda_k = e^{-\alpha|k|^2}$ for some parameter $\alpha > 0$) and let $\mathcal{J}$ be the index set defined by,

$$\mathcal{J} := \{(j_1, j_2) \in \mathbb{Z}^2 : -J_1/2 + 1 \leq j_1 \leq J_1/2, \ -J_2/2 + 1 \leq j_2 \leq J_2/2\}$$

The goal is to sample from the truncated expansion of $W(t)$,

$$W^J(t) = \sum_{j \in \mathcal{J}} \sqrt{\lambda_j} \phi_j \beta_j(t),$$

at the collection of sample points,

$$x_k = (L_1 k_1/J_1, L_2 k_2/J_2)^T, \qquad 0 \leq k_1 \leq J_1 - 1, \ 0 \leq k_2 \leq J_2 - 1.$$

Consider the random variable $Z(t_n, x)$ defined by,

$$Z(t_n, x) = \sqrt{\Delta t} \sum_{j \in \mathcal{J}} \sqrt{\lambda_j} \phi_j(x) \xi_j^n, \qquad \xi_j^n \sim \mathbb{CN}(0, 2),$$

meaning that $\xi_j^n = a + ib$ with $a, b \overset{\text{i.i.d}}{\sim} \mathcal{N}(0, 1)$ such that $Z(t_n, x_k)$ is a complex random variable with independent real and imaginary part with the same distribution as two independent copies of the increment $W^J(t_n + \Delta t, x_k) - W^J(t_n, x_k)$. Furthermore, $Z(t_n, x_k)$ can be expressed in the form,

$$Z(t_n, x_k) = \frac{1}{J_1 J_2} \sum_{j_1 = -J_1/2 + 1}^{J_1/2} \sum_{j_2 = -J_2/2 + 1}^{J_2/2} \widetilde{Z}_{j_1, j_2} e^{2i\pi\left(j_1 \frac{k_1}{J_1} + j_2 \frac{k_2}{J_2}\right)} \tag{7}$$

where $\widetilde{Z}_{j_1, j_2} = \sqrt{\Delta t \lambda_{j_1, j_2}} J_1 J_2 \xi_{j_1, j_2}^n$ We recognize that the matrix with entries given by eq. (7) is the 2D inverse DFT of the $J_1 \times J_2$ matrix with entries $\widetilde{Z}_{j_1, j_2}$. Therefore, we can sample two independent copies of

$$W^J(t_n + \Delta t, x_k) - W^J(t_n, x_k), \ 0 \leq k_1 \leq J_1 - 1, \ 0 \leq k_2 \leq J_2 - 1$$

by computing a single 2D inverse DFT.

### A.2.2 Cylindrical Wiener process

If the operator $Q = I$ is the identity, then $Q$ is not of trace class on $H$ so that the series in eq. (6) does not converge in $L^2(\Omega, H)$. This motivates the definition of cylindrical Wiener processes.

**Definition A.5 (Cylindrical Wiener process).** Let $H$ be a separable Hilbert space. A cylindrical Wiener process (a.k.a space-time white noise) is a $H$-valued stochastic process $\{W(t) : t \geq 0\}$ defined by

$$W(t) = \sum_{j=1}^{\infty} \phi_j \beta_j(t) \tag{8}$$

where $\{\phi_j\}$ is any orthonormal basis of $H$ and $\beta_j(t)$ are i.i.d. Brownian motions.

In all examples except Navier-Stokes, we drive the SPDE by samples $\xi$ from a cylindrical Wiener process in one dimension. Let $D = (0, L)$ and consider an $L^2(D)$-valued cylindrical Wiener process $W(t)$. As explained in Lord et al. [27, Example 10.31], if we take the basis

$$\phi_k(x) = \sqrt{2/L} \sin(k\pi x/L)$$

numerical approximation of sample paths from $W(t)$ are easy to obtain. The goal is to sample from the truncated expansion,

$$W^J(t) = \sum_{j=1}^{J} \phi_j \beta_j(t), \tag{9}$$

at the collection of sample points $x_k = kL/J$ for $k = 1, \ldots, J$. Observing that a trigonometric identity yields,

$$\text{Cov}\left(W^J(t, x_i), W^J(t, x_k)\right) = (tL/J)\delta_{ik}, \qquad i, k = 1, \ldots, J$$

the increments $W^J(t_n + \Delta t, x_k) - W^J(t_n, x_k) \sim \mathcal{N}(0, \Delta t L/J)$ for all $k = 1, \ldots, k$.

## A.3 Numerical solvers

In this section we present an overview of the numerical solvers for SPDEs we used to generate the data for all the experiments. The stochastic Ginzburg-Landau (Sec. 4.1 and appendix B.3), stochastic wave (Appendix B.4) equations have been solved using the finite difference method, while the stochastic Korteweg–De Vries (Sec. 4.2) and Navier Stokes (Sec. 4.3) equations have been solved using the spectral Galerkin method. We use the same setup as in Sec. 2. In particular, we focus on stochastic semilinear evolution equations of the form

$$du_t = (\mathcal{L}u_t + F(u_t)) \, dt + G(u_t) dW_t \tag{10}$$

where $W_t$ is either a $Q$-Wiener process or a cylindrical Wiener process and $\mathcal{L}$ is a linear differential operator generating a semigroup $e^{t\mathcal{L}}$. We consider nonlinearities $F, G$ regular enough (see Lord et al. [27, Assumption 10.23]) to guarantee existence and uniqueness of mild solutions of eq. (10) [27, Thm. 10.26].

### A.3.1 Finite difference method

We illustrate this numerical method for the reaction-diffusion equation

$$du_t = \left(\epsilon \partial_{xx}^2 u + F(u_t)\right) dt + \sigma dW_t, \quad u(0, x) = u_0(x),$$

with homogeneous Dirichlet boundary conditions and where $\epsilon, \sigma > 0$ are constants. We assume for simplicity that $u_0, u_t, W_t$ are real-valued and $\mathcal{D} = (0, a)$. The generalization to higher dimensions is straightforward.

Consider the grid points $x_j = jh$, where $h = \frac{a}{J}$ and $j = 0, \ldots, J$, for some spatial resolution $J \in \mathbb{N}$. Let $u_J(t)$ be the *finite difference approximation* of $[u(t, x_1), \ldots, u(t, x_{J-1})]$ (similarly for $W_J(t)$) resulting from the solution of the following SDE

$$du_J(t) = [-\epsilon M u_J(t) + \hat{f}(u_J(t))]dt + \sigma dW_J(t)$$

where $\hat{f}(u_J) = [f(u_1), \ldots, f(u_{J-1})]^T$ and $M$ is the $(J-1) \times (J-1)$ matrix approximating Laplacian (with free boundary conditions) which is given by

$$M = \frac{1}{h^2} \begin{pmatrix} 2 & -1 & & & & \\ -1 & 2 & -1 & & & \\ & -1 & 2 & -1 & & \\ & & \ddots & \ddots & \ddots & \\ & & & -1 & 2 & -1 \\ & & & & -1 & 2 \end{pmatrix}$$

One could modify $M$ for specific boundary conditions. For instance in the case of periodic boundary one should modify $M_{1,J-1} = M_{J-1,1} = -1$ (see Lord et al. [27, Chapter 3.4] for Dirichlet and

Neuman boundary condition modifications of $M$). To discretize in time, we may apply numerical methods for SDEs (see for example Lord et al. [27, Chapter 8]). Choosing the standard Euler-Marayama scheme with time step $\Delta t$ yields an approximation $u_{J,n}$ to $u_J(t_n)$ at $t_n = n\Delta t$ defined by

$$u_{J,n+1} = (I + \Delta t \epsilon M)^{-1} \left( u_{J,n} + \hat{f}(u_{J,n})\Delta t + \sigma(W_J(t_{n+1}) - W_J(t_n)) \right)$$

The increments $(W_J(t_{n+1}) - W_J(t_n))$ are generated using techniques discussed in Appendix A.2.

### A.3.2 Spectral Galerkin method

Consider again a separable Hilbert space $H$. Assume that the differential operator $\mathcal{L}$ in eq. (10) has a complete set of orthonormal eigenfunctions $\{\phi_j\}_{j\in\mathbb{N}}$ and eigenvalues $\lambda_j < 0$, ordered so that $\lambda_{j+1} < \lambda_j$. Then, we can define the semigroup $e^{t\mathcal{L}}$ as follows

$$e^{t\mathcal{L}}h = \sum_{j=1}^{\infty} e^{\lambda_j t} \langle h, \phi_j \rangle \phi_j, \quad h \in H.$$

Define the *Galerkin subspace* $V_J = \text{Span}\{\phi_1, ..., \phi_J\}$ and the orthonormal projections $P_J : H \to V_J$ as follows

$$P_J h = \sum_{i=1}^{J} \langle u, \phi_j \rangle \phi_j, \quad h \in H.$$

Then, the following defines *spectral Galerkin approximation* of eq. (10)

$$du_J(t) = (\mathcal{L}_J u_J(t) + P_J F(u_J(t)))dt + P_J G(u_J(t))dW_J(t), \quad u_J(0) = P_J u_0$$

where $u_J := P_J u$ and $\mathcal{L}_J := P_J \mathcal{L}$ and $W_J = P_J W$ is as in (9). Using a Euluer-Marayama discretization as above, we obtain the following discretization

$$u_{J,n+1} = (I + \Delta t \mathcal{L}_J)^{-1}(u_{J,n} + \Delta t P_J F(u_{J,n}) + P_J G(u_{J,n})\Delta W_{J,n}).$$

This approach is particularly convenient for problems with additive noise where the eigenfunctions of $\mathcal{L}$ and $Q$ (the covariance of the $Q$-Wiener process $W$) are equal, which is the case for all the experiments in this paper generated with this method. The eigenfunctions of the Laplacian with periodic boundary conditions correspond to the Fourier basis exponentials; therefore, one can define the projection $P_J$ in terms of the DFT.

## B    Further experiments

In this section with discuss additional details about the NSPDE model, its training procedure an of the baseline models, including how the relevant hyperparameters have been selected for each model.

### B.1    Derivation of the ODE parameterisation

If one assumes that $\mathcal{L}$ is a polynomial differential operator of degree $N$ of the form

$$\mathcal{L} = \sum_{n=0}^{N} \sum_{\substack{n_1,...,n_d \\ n_1+...+n_d=n}} C_{n_1,...,n_d} \frac{\partial^n}{\partial x_1^{n_1}...\partial x_d^{n_d}},$$

where $C_{n_1,...,n_d} \in \mathbb{C}^{d_h \times d_h}$ are complex matrices, then the FT of the kernel associated to $\mathcal{L}$ satisfies

$$\mathcal{F}(\mathcal{K}_t)(y) = e^{tP(iy)} \in \mathbb{C}^{d_h \times d_h},$$

for any $y \in \mathbb{C}^d$, where $e$ is the matrix exponential and $P$ is the following matrix-valued polynomial

$$P(y) = \sum_{n=0}^{N} \sum_{\substack{n_1,...,n_d \\ n_1+...+n_d=n}} (2\pi)^n y_1^{k_1}...y_d^{k_d} C_{n_1,...,n_d}.$$

Therefore, there exists a map $A : \mathbb{C}^d \to \mathbb{C}^{d_h \times d_h}$ such that $\mathcal{F}(\mathcal{K}_t)(y) = e^{tA(y)}$. It follows that

$$z_t = \mathcal{F}^{-1}\left(e^{tA}\mathcal{F}(z_0) + \int_0^t e^{(t-s)A}\mathcal{F}(H_{\theta,\xi}(z_s))ds\right) = \mathcal{F}^{-1}(v_t),$$

where $v_t : \mathbb{C}^d \to \mathbb{C}^{d_h}$ is the solution of the following ODE

$$v_t = v_0 + \int_0^t Av_s + \mathcal{F}(H_{\theta,\xi}(\mathcal{F}^{-1}(v_s))).$$

as shown in section 3.2.

## B.2   Additional experimental details

For all experiments the dataset is split into a training, validation and test sets with relative sizes $70\%/15\%/15\%$. For all models, a grid search on the hyperparameters is performed using the training and validation sets. We use the Adam optimizer and a scheduler which reads the validation loss and reduces the learning rate if no improvement is seen for a *patience* number of epochs. Additionally, an early stopping method is used to halt the training of the model if no improvement is seen after a *patience* number of epochs. The hyperparameters included in the grid search are stated below and examples of hyperparameter selection results are provided in tables 5 to 9.

**NSPDE**   The hyperparameters included in the grid search are the number of frequency modes used to parametrize the kernel in Fourier space $B = \mathcal{F}_{d+1}(\mathcal{K})$ and the number of forward iterations used to solve the fixed point problem.

**FNO**   The hyperparameters included in the grid search are the number of frequency modes used to parametrize the kernel and the number of layers $M$. Note that the numbers of frequency modes in the grid search differ from the ones used for the NSPDE model by a factor 2 to ensure that the effective number of retained modes is the same. For both the NSPDE model and FNO, we kept the number of hidden channels fixed to $d_h = 32$ as this systematically yielded better performances than previously included values and enabled to perform the grid search in a reasonable time.

**DeepONet**   The *Deep Operator Network* (DeepONet) [28] is another popular class of neural network models for learning operators on function spaces. The DeepONet architecture is based on the universal approximation theorem of Chen and Chen [6]. It consists of two sub-networks referred to as the *branch* and the *trunk* networks. The trunk acts on the coordinates $(t, x) \in [0, T] \times \mathcal{D}$, while the branch acts on the evaluation of the initial condition $u_0$ on a discretized grid $D$. Therefore, the DeepONet is not a space resolution-invariant architecture. The output of the network is expressed as

$$\text{DeepONet}(u_0)(t, x) = \sum_{k=1}^{p} b_k(u_0)\tau_k(t, x) + b_0,$$

where the $b_k$ and the $\tau_k$ are the outputs of the branch and trunk network respectively. The trunk network is usually a feedforward neural network, and one can chose the architecture of the branch network depending on the structure of the input domain. We follow Lu et al. [28] and use feedforward neural networks for both the trunk and the branch networks. We perform a grid search on the depth and width of the trunk and branch feedforward neural networks.

**NRDE/NCDE/NCDE-FNO**   The hyperparameters included in the grid search are the number of hidden channels and the type of solver as implemented by torchdiffeq [5]. We note that we used a depth-2 NRDE model (depth-2 already results in $d_\xi = 8\,385$ for forcings observed at 128 spatial points and higher depths models could not fit in memory) and recall that NCDE is a depth-1 NRDE.

| Table 5: Grid search NCDE (KdV) | | | | Table 6: Grid search NCDE-FNO (KdV) | | | |
|---|---|---|---|---|---|---|---|
| $d_h$ | # parameters | solver | validation loss | $d_h$ | # parameters | solver | validation loss |
| 8 | 136464 | rk4 | 0.511 | 8 | 5761 | rk4 | 0.140 |
| 16 | 272672 | rk4 | 0.510 | 16 | 15617 | rk4 | 0.142 |
| 32 | 545088 | rk4 | 0.505 | 32 | 48769 | rk4 | 0.145 |
| 8 | 136464 | euler | 0.560 | 8 | 5761 | euler | 0.310 |
| 16 | 272672 | euler | 0.561 | 16 | 15617 | euler | 0.314 |
| 32 | 545088 | euler | 0.556 | 32 | 48769 | euler | 0.321 |

Table 7: Grid search DeepONet (KdV)

| Branch & trunk width | Branch depth | Trunk depth | # parameters | validation loss |
|---|---|---|---|---|
| 256 | 3 | 4 | 1935616 | 0.258 |
| 128 | 3 | 4 | 885888 | 0.269 |
| 128 | 3 | 3 | 869376 | 0.279 |
| 128 | 3 | 2 | 852864 | 0.284 |
| 512 | 3 | 4 | 4526592 | 0.294 |
| 512 | 2 | 2 | 3738624 | 0.295 |
| 256 | 4 | 4 | 2001408 | 0.295 |
| 512 | 4 | 4 | 4789248 | 0.302 |
| 256 | 2 | 2 | 1738240 | 0.304 |
| 256 | 3 | 3 | 1869824 | 0.307 |
| 128 | 4 | 3 | 885888 | 0.311 |
| 128 | 4 | 2 | 869376 | 0.311 |
| 256 | 3 | 2 | 1804032 | 0.313 |
| 128 | 2 | 2 | 836352 | 0.315 |
| 256 | 4 | 2 | 1869824 | 0.316 |
| 128 | 4 | 4 | 902400 | 0.318 |
| 256 | 4 | 3 | 1935616 | 0.319 |
| 512 | 4 | 3 | 4526592 | 0.320 |
| 512 | 4 | 2 | 4263936 | 0.334 |
| 512 | 3 | 2 | 4001280 | 0.350 |
| 128 | 2 | 4 | 869376 | 0.350 |
| 512 | 3 | 3 | 4263936 | 0.359 |
| 512 | 2 | 3 | 4001280 | 0.366 |
| 256 | 2 | 3 | 1804032 | 0.370 |
| 128 | 2 | 3 | 852864 | 0.382 |
| 256 | 2 | 4 | 1869824 | 0.392 |
| 512 | 2 | 4 | 4263936 | 0.395 |

Table 8: Grid search FNO (KdV)

| $d_h$ | depth | modes 1 | modes 2 | # parameters | validation loss |
|---|---|---|---|---|---|
| 32 | 1 | 16 | 16 | 532993 | 0.163 |
| 32 | 1 | 16 | 50 | 1647105 | 0.117 |
| 32 | 1 | 32 | 16 | 1057281 | 0.163 |
| 32 | 1 | 32 | 50 | 3285505 | 0.117 |
| 32 | 2 | 16 | 16 | 1058337 | 0.163 |
| 32 | 2 | 16 | 50 | 3286561 | 0.120 |
| 32 | 2 | 32 | 16 | 2106913 | 0.163 |
| 32 | 2 | 32 | 50 | 6563361 | 0.118 |
| 32 | 3 | 16 | 16 | 1583681 | 0.163 |
| 32 | 3 | 16 | 50 | 4926017 | 0.120 |
| 32 | 3 | 32 | 16 | 3156545 | 0.163 |
| 32 | 3 | 32 | 50 | 9841217 | 0.120 |
| 32 | 4 | 16 | 16 | 2109025 | 0.163 |
| 32 | 4 | 16 | 50 | 6565473 | 0.122 |
| 32 | 4 | 32 | 16 | 4206177 | 0.163 |
| 32 | 4 | 32 | 50 | 13119073 | 0.120 |

Table 9: Grid search NSPDE (KdV)

| $d_h$ | Picard's iterations | modes 1 | modes 2 | # parameters | validation loss |
|---|---|---|---|---|---|
| 32 | 1 | 32 | 32 | 1055233 | 0.023 |
| 32 | 1 | 32 | 100 | 3283457 | 0.011 |
| 32 | 1 | 64 | 32 | 2103809 | 0.030 |
| 32 | 1 | 64 | 100 | 6560257 | 0.009 |
| 32 | 2 | 32 | 32 | 1055233 | 0.018 |
| 32 | 2 | 32 | 100 | 3283457 | 0.012 |
| 32 | 2 | 64 | 32 | 2103809 | 0.015 |
| 32 | 2 | 64 | 100 | 6560257 | 0.010 |
| 32 | 3 | 32 | 32 | 1055233 | 0.016 |
| 32 | 3 | 32 | 100 | 3283457 | 0.013 |
| 32 | 3 | 64 | 32 | 2103809 | 0.022 |
| 32 | 3 | 64 | 100 | 6560257 | 0.011 |
| 32 | 4 | 32 | 32 | 1055233 | 0.019 |
| 32 | 4 | 32 | 100 | 3283457 | 0.012 |
| 32 | 4 | 64 | 32 | 2103809 | 0.021 |
| 32 | 4 | 64 | 100 | 6560257 | 0.016 |

## B.3 Stochastic Ginzburg-Landau equation

Recall that the stochastic Ginzburg-Landau equations are of the form,

$$\partial_t u - \Delta u = 3u - u^3 + G(u)\xi, \tag{11}$$
$$u(0,x) = u_0(x), \quad (t,x) \in [0,T] \times [0,1]$$

subject to either Periodic or Dirichlet boundary conditions. Periodic boundary conditions are given by $u(t,0) = u(t,1)$ for all $t \geq 0$ and Dirichlet boundary conditions are given by $u(t,0) = u(t,1) = 0$ for all $t \geq 0$. Initial condition we take as in Sec. 4.1 $u_0(x) = x(1-x) + \kappa\eta(x)$ with $\kappa = 0$ or $\kappa = 0.1$ depending on a task. In both Periodic and Dirichlet case we can take $\eta(x)$ as in Sec. 4.1 though in Dirichlet case one must take $a_0 = 0$ to ensure $u_0$ being zero at the boundary.

We first reproduce an experiment from Sec. 4.1 on the additive stochastic Ginzburg-Landau equation but with Dirichlet boundary conditions instead of the periodic. We compare it to the benchmark of FNO model which was the most successful among all the benchmarks of Sec. 4. From Table 10 we

see that even though Neural SPDE model depends on the spectral methods the errors did not increase compared to the periodic equation in Sec. 4.1 (see Table 1). Our algorithm still outperforms FNO whose relative $L2$ error increased slightly. The fact that Neural SPDE can be applied to non-periodic equations could be perhaps explained by interpolation ($L_\theta$) and projection ($\Pi_\theta$) neural networks that could correct for non-periodicity of the data.

Table 10: **Additive stochastic Ginzburg-Landau equation with homogeneous Dirichlet boundary conditions**. The experimental setup is the same as in the main paper. We report the relative L2 error on the test set. The symbol x indicates that the model is not applicable. $N$ is fixed to $1\,000$.

| Model | $u_0 \mapsto u$ | $\xi \mapsto u$ | $(u_0, \xi) \mapsto u$ |
|---|---|---|---|
| FNO | 0.132 | 0.023 | x |
| NSPDE (Ours) | 0.135 | 0.008 | 0.010 |

We now take a look at the specific hyperparameter: number of forward iterations in the fixed point solver. We also call this a number of Picard iterations $P$. Theoretically as $P$ increases Fixed Point Solver should converge to the true solution (see [11]). This suggests that higher $P$ should improve the performance of the Neural SPDE algorithms. In practise we observed in both additive Ginsburg Landau equation from Sec. 4.1 and in KdV equation from Sec. 4.2 that $P = 1$ could already be enough. This could be explained either by dominance of the linear part of the equation or by overfitting in these cases. Thus we present an experiment on the multiplicative stochastic Ginzbug-Landau equation over a longer (compared to Sec. 4.1) time interval. In the Table 11 we compare NSPDE with $P \in \{1, 2, 3, 4\}$ and again include FNO benchmark (which performed best in the previous experiments). We see that NSPDE with even $P = 1$ outperforms FNO. Relative $L2$ error for $T = 0.05$ increases for both NSPDE and FNO due to more complicated multiplicative noise. In Table 11 we present for each $P$ the best result over other hyperparameters obtained by cross validation. One could clearly see an improvement in error as we increase the number of Picard iterations $P$ (with an exception of the case $T = 0.05$ where $P = 3$ outperformed $P = 4$). This improvement becomes more apparent as the time frame $T$ increases. Heuristically (and qualitatively) this is due to the fact that for the short times solution of the SPDE is relatively close to its linearised version and that nonlinearity of the equation starts to play a bigger role for larger $T$.

Table 11: **Multiplicative stochastic Ginzburg-Landau equation**. We report the relative L2 error on the test for FNO and NSPDE (Ours) for different number of Picard iterations on the task $\xi \to u$.

| Time horizon | FNO | NSPDE ($P = 1$) | NSPDE ($P = 2$) | NSPDE ($P = 3$) | NSPDE ($P = 4$) |
|---|---|---|---|---|---|
| $T = 0.05$ | 0.040 | 0.023 | 0.018 | **0.016** | 0.017 |
| $T = 0.10$ | 0.068 | 0.042 | 0.041 | **0.040** | **0.040** |
| $T = 0.25$ | 0.105 | 0.079 | 0.077 | 0.073 | **0.072** |

### B.4   The stochastic wave equation

In this section we consider the following nonlinear wave equation with multiplicative stochastic forcing,

$$\partial_t^2 u - \Delta u = \cos(\pi u) + u^2 + u\xi, \tag{12}$$
$$u(t, 0) = u(t, 1),$$
$$u(0, x) = u_0(x),$$
$$\partial_t u(0, x) = v_0(x), \quad (t, x) \in [0, T] \times [0, 1].$$

The nonlinear stochastic wave equation arises in relativistic quantum mechanics and is also used in simulations of nonlinear waves that are subject to either noisy observations or random forcing. We refer a reader to Temam [34] for an overview on the nonlinear wave equation. The above equation can put in a form of eq. (2) by rewriting it as a system for $(u, v) = (u, \partial_t u)$. To generate training

datasets, we solve the SPDE using a finite difference method with 128 evenly distanced points in space and a time step size $\Delta t = 10^{-3}$. As in Chevyrev et al. [7, eq. (3.5)], we solve the SPDE until $T = 0.5$. We then downsample the temporal resolution by a factor 5, resulting in 100 time points. Here, the initial condition is given by $u_0(x) = \sin(2\pi x) + \kappa\eta(x)$, where $\eta$ is defined in Sec. 4.1 and for simplicity initial velocity $v_0$ is taken deterministic $v_0(x) = x(1-x)$. Similarly to Sec. 4.1 we either take $\kappa = 0$ or $\kappa = 1$ to generate datasets where the initial condition is either fixed or varies across samples. Each dataset consists of $N = 1\,000$ training observations.

Table 12: **Stochastic Wave equation**. We report the relative L2 error on the test set. The symbol x indicates that the model is not applicable. $N$ is fixed to $1\,000$.

| Model | $u_0 \mapsto u$ | $\xi \mapsto u$ | $(u_0, \xi) \mapsto u$ |
|---|---|---|---|
| NCDE | x | 0.142 | 0.432 |
| NRDE | x | 0.146 | 0.445 |
| NCDE-FNO | x | 0.029 | 0.037 |
| DeepONet | 0.190 | 0.143 | x |
| FNO | 0.151 | 0.026 | x |
| NSPDE (Ours) | 0.150 | **0.023** | **0.026** |

## B.5 Deterministic Navier-Stokes PDE

In this final experiment, we demonstrate that our Neural SPDE model can also be used in the setting of PDEs without any stochastic term. We do so by studying the example from [25] on deterministic Navier-Stokes. More precisely, we consider the 2D Navier-Stokes equation for a viscous, incompressible fluid in vorticity form:

$$\partial_t w(t,x) - \nu\Delta w(t,x) = f(x) - u(t,x) \cdot \nabla w(t,x), \qquad t \in [0,T], x \in [0,1]^2 \qquad (13)$$

$$\nabla \cdot u(t,x) = 0, \qquad t \in [0,T], x \in [0,1]^2 \qquad (14)$$

$$w(x,0) = w_0(x), \qquad x \in [0,1]^2 \qquad (15)$$

where $u : [0,T] \times [0,1]^2 \to \mathbb{R}^2$ is the velocity field, $w = \nabla \times u$ is the vorticity with $w_0 : [0,1]^2 \to \mathbb{R}$ being the initial vorticity. Here $f$ is a deterministic forcing term which we take as in [25]. We follow the experimental setup from [25] and use the dataset (available under an MIT license) where $\nu = 10^{-5}$, $N = 1000$ and $T = 20$. We achieve similar performances as FNO with a L2 error of 0.17. A comparison between a true and predicted trajectory is depicted in Fig. 3.

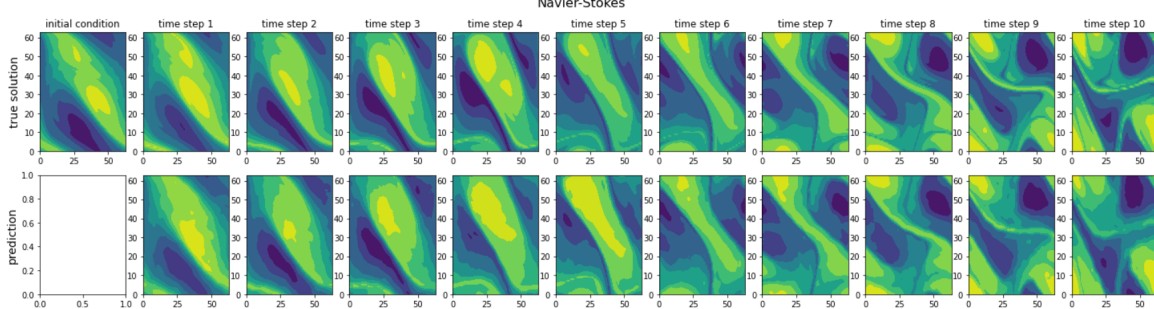

Figure 3: **Top panel:** Initial vorticity and ground truth vorticity at later time steps on a $64 \times 64$ mesh. **Bottom panel:** Predictions of the Neural SPDE model.