# OpenReview forum: "Neural Stochastic PDEs: Resolution-Invariant Learning of Continuous Spatiotemporal Dynamics"
_NeurIPS.cc/2022/Conference — NeurIPS 2022 Accept_

### Official Review · Reviewer_98nh · 2022-07-10

**Rating:** 7
**Confidence:** 3
**Soundness:** 2 fair
**Presentation:** 2 fair
**Contribution:** 3 good

**Summary:**

- The paper introduces Neural Stochastic Partial Differential Equations (NSPDEs) for learning solution operators for SPDEs while being resolution invariant.
- Two methods of performing inference for NSPDEs are proposed: (1) A method using Fourier Transformations to reduce the NSPDE to a Neural ODE, which can then be solved with existing techniques; (2) a method based on framing the NSPDE as a fixed point problem allowing the use of fixed-point solvers.
- The model is evaluated on well-known SPDEs and shown to outperform various Neural Differential Equation models as well as the Neural Operators method.

**Questions:**

- See all my questions from the weaknesses section above.
- I am a bit confused about how $\xi(t, x)$ is supplied to the Neural Operator models and all models more broadly when it is observed.
   - First of all, what do we assume it is observed more precisely? Do we have full access to the function $\xi(t, x)$ and we can sample it at any point or do we have access to some samples from this function for certain $(t, x)$ pairs.
   - In the case of FNO, in its original paper, the model takes as input a function over the domain $\mathcal{D}$ and outputs another function over the same domain. As far as I understand, in the setting when $u_0$ is fixed, the input function to the FNO model becomes $\xi(t, x)$. So you are supplying to, FNO samples from $\xi(t, x)$ on a regular grid over $[0, T] \times \mathcal{D}$. Is that correct?
   - I see that the Neural Operator models were not evaluated in the setting where the input is $(u_0, \xi)$. Couldn't this input be supplyed as function combining these two functions? For instance, the new function could concatenate the two over the spatial domain at $t = 0$ and concatenate $\xi$ with some dummy values for $t > 0$ (or something else along these lines)?


**Ethics Review Area:**

["I don’t know"]

**Limitations:**

TLDR: No
- For limitations, the checklist mentions Section 2 (which is a background section), while Section 5 discusses future directions (not limitations). So I believe limitations have not been properly addressed.
- Negative societal impacts are not discussed either. I believe this part is treated too superficially despite that fact that PDEs show up everywhere in mechanics and these sort of models could be applied for all sorts of military applications and what not. Therefore, I would invite the authors to put more thought about the potentially dangerous applications of these models.

**Strengths And Weaknesses:**

# Stengths

- The paper proposes a well-executed instantiation of Neural SPDEs continuing the recent research direction of combining deep learning models with various types of differential equations.
- The two proposed methods for evaluating the NSPDE model provide some degree of choice/flexibility for practitioners.
- The emprical evaluation shows that the proposed model performs well at predicting the solution of SPDEs.

# Weaknesses

- SPDEs are notoriously a very technical subject, however, I believe this is not a justification for the fact that the background section fails (and perhaps does not even attempt) to find the right level of technicality for a machine learning audience. Overall, the background section is not self-contained at all:
     - The reader is invited to consult a primer on semigroup theory. But no semigroup theory is actually needed to read the main content of the paper. The word semigroup is used only to say that $\mathcal{L}$ is a differential operator generating a semigroup $e^{t\mathcal{L}}$. Why not simply say $\mathcal{L}$ induces a map $e^{t\mathcal{L}}$ and refer to it as a map. If you really want to mention it's a semigroup, you can add a footnote and even define what a semigroup is in that footnote (it's a very short definition).
     - Line 115 mentions a "mollifier" without ever explaining what that is. Also, what does $\delta^\epsilon * W$ mean? Is that a convolution?
     - This is followed by a very technical discussion in Lines 116-130 that is of little consequence for understanding the rest of the paper. This discussion could be perhaps moved to the appendix. The space of that paragraph could be saved for further explaining other concepts or adding some figures that help explaining the main concepts.
     - For Wiener processes, the paper simply references definitions from papers/books. I understand these have been defined properly in the appendix, but I think a short definition or at least an intuitive definition would have been possible in the main text given they are the essence of SPDEs. This would make the main text more self-contained.
- There is no comparison between the two evaluation methods. First, what is their computational complexity and what are their theoretical advantages and drawbacks? Second, how do they compare in practice / empirically? None of these two types of questions is addressed and from reading the paper, it is unclear which one of the two one should use in various settings. Instead, a minimal set of guidelines backed up by theory and experiments could be provided.
- None of the tables contain error bars. I see that the checklist says this is because "this would be too expensive for the baseline models NCDE, NRDE and NCDE-FNO". First, the neural operator models do not contain error bars either despite being faster. Second, this claim is not really backed up by data. Could you report the training and evaluation time of those models that made evaluation across multiple seeds difficult/impossible? Realistically speaking, if there was enough compute to run these experiment once, then running them 3 or 5 times should be doable (unless it takes weeks or months?). At least on the systems with one spatial dimensions, I don't think that should be the case.
- One of the main motivations of the model is that it generalises NCDEs and it can also be applied to irregularly sampled and partially observed data. Correct me if I am wrong, but I believe that none of the experiments in the evaluation section actually consider such a context. Here, I would have expected something like in Figure 1, where data is observed at some sequence of times $t_1, \ldots, t_n$ and the samples might be missing some channels.

======== EDIT =========

Raised my score to Accept after discussion.

---

> ### Author Response · Authors · 2022-08-02
> **Response to Reviewer 98nh (1/3)**
>
> Thank you for the useful feedback and constructive criticisms. Please find below our responses to the raised concerns.
>
> **Background section**
>
> We agree that the definitions of semigroup and mollifier are rather simple and so we follow your suggestion and add both definitions as footnotes to the revised version of the paper (page 3, see attached), where we also specified that $*$ means indeed convolution. Concerning the definition of Wiener process, we believe it is challenging to incorporate the latter within the main text given the page limit. We note that we already mention in l.112-113 (of the previous version) that an SPDE can be intuitively thought of as an SDE driven by an infinite dimensional Brownian motion.
>
> We agree that the technical paragraph in l.116-130 (of the previous version) concerning convergence to a limiting SPDE is perhaps of limited interest for a machine learning audience, so we follow your suggestion and move this discussion to the appendix. We also agree that notation in Sections 3.1 and 3.2 can be made lighter by transferring some of the details on how to obtain the ODE parameterisation (l. 160-163) to the appendix and by merging equations in l. 180 and 182 into a single equation. These changes have been incorporated into the revised version of the paper (see attached).
>
> In addition, to address the concern about clarity of the method, we added a new schematic (Figure 1) to the revised version of the paper; we hope the latter will facilitate the understanding of how the model operates.
>
> **Comparison between evaluation methods**
>
> The concern regarding a comparison between the two evaluation methods has been raised also by reviewer QRvu, and we recognise it is a lacking aspect of our paper. Therefore, we took special care to address this issue by including a thorough comparison between the two evaluation methods in Section 3.5 of the updated version of the paper (see attached).
>
> We ran all experiments with both evaluation methods achieving comparable performance; we found that the ODE approach is approximately 10 times slower than the Fixed Point approach; this is the main reason why we chose not to include the former in the comparison. We believe that this similarity in performance is to be expected as both methods provide an approximation of the mild solution of the same SPDE in latent space.
>
> **Error bars**
>
> The main reason why we did not report error bars is because repeating the experiments 3 or 5 times would have caused a significant increase in the overall financial cost of the project as all experiments were performed on virtual machines rented on cloud services and with limited budget. However, we agree that the presence of error bars would be beneficial for showcasing the robustness in performance of our model across different runs. For this reason, we performed additional runs to obtain error bars on the Stochastic kDV example for both the FNO and NSPDE models and report the results in the table below.
>
> Model|$\xi\mapsto u$|$(u_0,\xi)\mapsto u$
> --- |--- |---
> FNO|0.079 (0.001)| $\quad $ x
> NSPDE (Ours) |0.004 (0.001) |0.008  (0.001)

---

> > ### Author Response · Authors · 2022-08-02
> > **Response to Reviewer 98nh (2/3)**
> >
> > **Irregular sampling and partial observations**
> >
> > Being a generalisation of NCDEs to infinite dimensional state spaces, NSPDEs are indeed able to process signals that are irregularly sampled both in space and in time by interpolating between observations.
> >
> > It is important to keep in mind though that to ensure good approximation of the FT by the FFT, the interpolation must translate the irregular data to a (possibly finer) regular grid.
> >
> > We would like to emphasise that the ability of a model to process irregular data does not guarantee its robustness when some observations are dropped. Such robustness can only be guaranteed if the original signal is smooth enough so that dropping observations and replacing them by interpolation results in a new signal that is close, in some suitable norm, to the original signal. This is indeed the case for the Character Trajectory example considered in the original NCDE paper ([1] sec. 4.1). However if the driving signal is less regular, as it is the case for Brownian motion in NSDEs, then dropping points at random and smoothly interpolating the irregular signal would completely change the dynamics and drastically deteriorate performance of the model.
> >
> > In some of our settings (stochastic KdV and Navier-Stokes), the driving signal is a Q-Wiener process, which is rough in time, but smoother in space. To back our claims, we run two additional experiments on the Stochastic KdV example where we dropped uniformly at random 1) 10% of the data in time and 2) 50% of the data in space. As it can be observed in the table below, the results confirm what mentioned above, and we add this table, as well as a discussion, to the revised version of the paper (last paragraph in Section 4.2).
> >
> > Subsampling rates | $\xi\mapsto u$ | $(u_0,\xi)\mapsto u$
> > ---|---|---
> >  Time: 0%, Space: 0% | 0.004| 0.008
> > Time: 10%, Space: 0% | 0.076| 0.059
> > Time: 0%, Space: 50% | 0.005 | 0.008
> >
> > For completeness, given that in our experiments we didn’t consider smooth driving signals (which would have allowed to showcase robustness of NSPDE also to irregular sampling also in time), we propose to replace the following sentence in the abstract:
> >
> > *…capable of processing incoming sequential information arriving irregularly in time and observed at arbitrary spatial resolutions…*
> >
> > by the new sentence
> >
> > *…capable of processing incoming sequential information observed at arbitrary spatial resolutions…*
> >
> > **Additional questions**
> >
> > Concerning the additional questions on how the noise is supplied to the models, please find our answers below inline.
> >
> > - **Q** *First of all, what do we assume it is observed more precisely? Do we have full access to the function  and we can sample it at any point or do we have access to some samples from this function for certain pairs*. **A** We have access only to the evaluation of $\xi$ on some given space-time grid.
> > - **Q** *In the case of FNO, in its original paper, the model takes as input a function over the domain  and outputs another function over the same domain. As far as I understand, in the setting when  is fixed, the input function to the FNO model becomes . So you are supplying to, FNO samples from  on a regular grid over . Is that correct?* **A** Yes.
> > - **Q** *I see that the Neural Operator models were not evaluated in the setting where the input is . Couldn't this input be supplyed as function combining these two functions? For instance, the new function could concatenate the two over the spatial domain at  and concatenate  with some dummy values for  (or something else along these lines)?* **A** As mentioned in our response to reviewer QRvu, we note that the Hilbert spaces of functions where $u_0$ (and $u$) and the noise $\xi$ are defined could in principle be completely different, therefore it is mathematically difficult to see how a FNO, as defined in the original paper, can consume both objects simultaneously. However, we agree that considering $u_0$ and $\xi$ as generic blocks of data, it is indeed possible to perform a handcrafted modification of these tensors, for example repeat $u_0$ across the time channel, concatenate this constant path to the path $\xi$ and then apply a FNO to this augmented data stream. We did try to do so, but the performance of the resulting model was extremely poor across all considered experiments.

---

> > > ### Author Response · Authors · 2022-08-02
> > > **Response to Reviewer 98nh (3/3)**
> > >
> > > **Limitations**
> > >
> > > We agree that the theoretical limitations discussed at the end of the paper, despite being interesting for a mathematical audience, are perhaps not so relevant for a machine learning audience, therefore we move these to the appendix in the revised version.
> > >
> > > Something we did not mention but realised since the submission is that similarly to other neural operator models, parameterising the kernel in Fourier space is by no means the only available option, see for example [2]. Other parameterisations could mitigate some of the disadvantages of the FFT  (irregular boundaries, aliasing effect etc.);  we add these considerations to a separate section with header **Limitations and future work** in the conclusion of the revised version of the paper.
> > >
> > > **References**
> > >
> > > [1] Kidger, P., Morrill, J., Foster, J., & Lyons, T. (2020). Neural controlled differential equations for irregular time series. Advances in Neural Information Processing Systems, 33, 6696-6707.
> > >
> > > [2] Li, Z., Kovachki, N., Azizzadenesheli, K., Liu, B., Bhattacharya, K., Stuart, A., & Anandkumar, A. (2020). Neural operator: Graph kernel network for partial differential equations. arXiv preprint arXiv:2003.03485.

---

> > ### Comment · Reviewer_98nh · 2022-08-05
> > **Response to Authors**
> >
> > I thank the authors for their ample response! I believe all my concerns (many also expressed by Reviewer QRvu) have been addressed. In particular, I welcome the changes to improve the readability of the background section and, overall, the entire paper.
> >
> > At the same time, I also shared the reviewer's QRvu concerns about the applicability of this model and the main problem it is tackling. Therefore, I have one further question about this to improve my understanding before deciding on a final score:
> > - Since (correct me if I am wrong) the realisation of the noise is not typically observed in real-world phenomena, is your model applicable to learning SPDE dynamics from real-world observations (where only the state at certain locations and times is observed)?
> > - As far as I understand, I think the main type of application this model is targeting would be speeding up complex simulations involving SPDEs and increasing their accuracy at arbitrary resolutions. This is because, in simulation (unlike the real-world), one can easily have access to realisations of the noise term. Do the authors share this view or do they have other applications in mind?

---

> > > ### Author Response · Authors · 2022-08-06
> > > **Response to Reviewer 98nh**
> > >
> > > Providing accurate and fast approximations to solution operators of SPDEs (mapping realisations of the input noise to the corresponding pathwise solutions) is a central challenge in many areas of numerical analysis such as **non-linear filtering** and **data assimilation**, with important applications ranging from weather predictions to the study of ocean dynamics [1,2,3]. As mentioned in the second paragraph of the introduction, to ensure accurate predictions with classical numerical solvers, computations must be performed on very fine discretization grids, often making the overall computational cost prohibitive. Being resolution-invariant and fast to evaluate, we believe NSPDEs provide a new class of models addressing the aforementioned limitations.
> > >
> > > We also note that a special class of NCDEs, known as NSDEs, has been used to build continuous-time generative models for time series [4]. The input Brownian trajectories are transformed by the NSDE to time-evolving paths that are matched to some observed data distribution by means of a discriminator on pathspace. Analogously, NSPDEs can serve as natural **generators for real-world spatiotemporal signals** where only the state at certain locations and times is observed, and where realisations of the noise are not observed but generated from a reference process (e.g. Wiener processes). We find of particular interest the definition and computation of an appropriate discrepancy for probability measures supported on spatiotemporal signals, an application we mentioned as future research direction in the paper and that we are currently investigating.
> > >
> > > Finally, despite also being inspired from differential equations, NCDEs have been predominantly used outside the realm of numerical analysis to tackle tasks such as time series classification and prediction. NSPDEs can be thought as a **continuous-time-space analogue of CNN-RNN** models, as NCDEs are to RNNs and as Neural ODEs are to ResNets [5]. In this paper, we focus on solving SPDEs, where the relevant input signals are sample paths from Wiener processes. However, and in complete analogy with NCDEs, we expect NSPDEs to be deployed in other contexts, for example in computer vision to process videos at arbitrary resolution.
> > >
> > > **References**
> > >
> > > [1] Crisan, D., Holm, D. D., Luesink, E., Mensah, P. R., & Pan, W. (2021). Theoretical and computational analysis of the thermal quasi-geostrophic model. arXiv preprint arXiv:2106.14850.
> > >
> > > [2] Cotter, C., Crisan, D., Holm, D. D., Pan, W., & Shevchenko, I. (2019). Numerically modeling stochastic Lie transport in fluid dynamics. Multiscale Modeling & Simulation, 17(1), 192-232.
> > >
> > > [3] X. Shi, Z. Chen, H. Wang, D.-Y. Yeung, W.-k. Wong, and W.-c. Woo. Convolutional lstm network: A machine learning approach for precipitation nowcasting. In Proceedings of the 28th International Conference on Neural Information Processing Systems - Volume 1, NIPS’15, 802–810. MIT Press, Cambridge, MA, USA, 2015.
> > >
> > > [4] Kidger, P., Foster, J., Li, X., & Lyons, T. J. (2021, July). Neural sdes as infinite-dimensional gans. In International Conference on Machine Learning (pp. 5453-5463). PMLR.
> > >
> > > [5] Wang, J., Yang, Y., Mao, J., Huang, Z., Huang, C., & Xu, W. (2016). Cnn-rnn: A unified framework for multi-label image classification. In Proceedings of the IEEE conference on computer vision and pattern recognition (pp. 2285-2294).

---

> > > > ### Comment · Reviewer_98nh · 2022-08-08
> > > > **Response**
> > > >
> > > > Dear authors,
> > > >
> > > > I did read the response you already provided to Reviewer QRvu so there was no need to copy-paste it here. My question was a bit more specific and related to the fact that almost all Neural DE models are applicable to learning dynamics from real-world observations from some unknown dynamics. However, when the dynamics are stochastic one does not have access to the noise observations in the real world so I was asking if your model is still useful in those sorts of applications (say compared to FNO). Perhaps scenarios where the noise is not observed can be discussed as part of the limitations of the model.
> > > >
> > > > In any case, I understand that the main application the paper is targeting is fast and accurate numerical simulations that also generalise to other resolutions (unlike solvers). I believe that in this regard the paper has provided sufficient experimental evidence that the proposed model performs well. The authors have also provided during this discussion other potentially promising applications that might prove fruitful in future work. Therefore, I will also increase my score to accept.

---

### Official Review · Reviewer_QRvu · 2022-07-11

**Rating:** 6
**Confidence:** 2
**Soundness:** 3 good
**Presentation:** 2 fair
**Contribution:** 2 fair

**Summary:**

This paper introduces a method to elucidate noise-conditioned spatiotemporal dynamics via the framework of Stochastic Partial Differential Equations (SPDEs). It consists in learning neural-parameterized SPDEs to fit the observed phenomenon thanks to the adaptation of two standard Fourier-based resolution methods used to handle the neural SPDEs (an ODE solver in the Fourier space and a fixed-point method). This allows the proposed model to work on arbitrary space and time resolutions.

The model is then evaluated against several neural operators and neural CDE baselines to solve standard SPDEs in three contexts: based on the initial condition only, conditioned on the intrinsic noise information, and both at the same time. The model especially shows significant improvement w.r.t. the considered baselines on the two latter settings.

**Questions:**

 - Could the authors comment in the ability of the model to handle partially observed stochasticity and the applicability of the considered setting to real-world phenomena?
 - Can the experimental setting be clarified w.r.t. my concerns described above?

**Limitations:**

The authors adequately addressed the limitations of their work.

**Strengths And Weaknesses:**

### Strengths

The paper presents, to my knowledge, a **well-motivated, novel model** for a **relevant task**, since handling an underlying noise in the observed dynamics remains a challenge for neural PDE-solving methods. **Numerical results support the relevance** of the proposed model, with the latter being able to accurately predict the studied phenomena, unlike tested baselines. Its **flexibility** is also appreciated, with its two different versions, its ability to handle multiple output resolutions and its versatility in the experiments.

### Weaknesses

I have three main concerns about this paper that are significant enough to nuance my positive recommendation.

Firstly, **the significance of both the model and its results remains unclear**. This is in part due to its SPDE inspiration: the model is specifically designed to integrate the complete noise information in its predictions, but this use case seems limited. Indeed, accessing this noise may only be possible in synthetic controlled settings such as the conducted experiments in the paper. Assessing whether the model is able to integrate partially observed noise would greatly benefit the paper in this regard. Nonetheless, given the underperformance of the model in the absence of noise information, I suspect that this may be challenging for the model as currently presented here.

Secondly, I find that **the evaluation is lacking in some respects**.
 - Some baselines on neural operators might be missing, in particular improvements of FNO: for example, the works of Cao (2021) and Gupta et al. (2021) seem relevant. The evaluation of FNO is also questionable: why is it necessarily unable to handle the $(u_0, \xi) \rightarrow u$ case? Couldn't $u_0$ be appended to $\xi$ at any $t$ as a supplementary information?
 - Moreover, the impact of the two proposed resolution methods is not evaluated: an ablation study (in terms of performance, efficiency, etc.) might highlight their respective merits w.r.t. other standard resolution methods. On this matter, to my understanding, only one of these alternatives (the fixed-point one) is used in the experiments.
 - Finally, I am unsure of the relevance of the task $u_0 \rightarrow u$ in which models are trained and evaluated using a regression objective while the phenomenon is stochastic by nature.

Thirdly, **the writing and presentation of the method should be improved**. The problem is threefold.
 1. The SPDEs are contextualized by a dense background description which is hard to follow for non-experts. Providing more intuitions and simplifying the notations could improve this point: for example, the simplification of lines 148-149 could be done earlier in the document, and examples accompanying the introduction of SPDEs would be welcome.
 2. The presentation of the method in unclear on some points, notably because of the heavy notations but also because important information is scattered across the paper. In particular, Section 3 should clearly state the possible inputs and outputs of the model as well as the learnable parameters (for example, the fact that "Neural SPDE learns a representation of $\mathcal{G}$" is only explicitly stated in Section 4, line 235).
 3. The paper misses a further discussion of related work, either by adding a dedicated section or by specifying in e.g. Section 3 the precise differences with previous work. This would make the original contributions of the paper clearer.

Cao. Choose a Transformer: Fourier or Galerkin. NeurIPS 2021.\
Gupta et al. Multiwavelet-based Operator Learning for Differential Equations. NeurIPS 2021.

### Overall Sentiment

Despite its flaws which are not prohibitive, I believe that this paper introduces an interesting method for the NeurIPS community with appealing performance. Therefore, I am inclined to recommend this paper for acceptance. I am willing to change my evaluation following the discussion phase with the authors and the other reviewers.

**Post-rebuttal update:** The authors' response partially alleviated my concerns (cf. [my answer below](https://openreview.net/forum?id=OGM9dXemmq&noteId=8iHe2W6IV1B)). I consequently raise my score from 5 to 6.

---

> ### Author Response · Authors · 2022-08-02
> **Response to Reviewer QRvu (1/2)**
>
> Thank you for the useful feedback and constructive criticisms. Please find below our responses to the raised concerns.
>
> **Significance of model and results**
>
> Providing accurate and fast approximations to solution operators of SPDEs (mapping realisations of the input noise to the corresponding pathwise solutions) is a central challenge in many areas of numerical analysis such as non-linear filtering and data assimilation, with important applications ranging from weather predictions to the study of ocean dynamics [1,2,3]. As mentioned in the second paragraph of the introduction, to ensure accurate predictions with classical numerical solvers, computations must be performed on very fine discretization grids, often making the overall computational cost prohibitive. Being resolution-invariant and fast to evaluate, we believe NSPDEs provide a new class of models addressing the aforementioned limitations.
>
> More generally, NSPDEs can be thought of as an extension of NCDEs to cases where the input and/or output signals evolve in some function spaces. Despite also being inspired from differential equations, NCDEs have been predominantly used outside the realm of numerical analysis to tackle tasks such as time series classification and prediction. In this paper, we focus on solving SPDEs, where the relevant input signals are sample paths from Wiener processes. However, and in complete analogy with NCDEs, we expect NSPDEs to be deployed in other contexts, for example in computer vision to process videos at arbitrary resolution.
>
> We also note that a special class of NCDEs, known as NSDEs, has been used to build continuous-time generative models for time series [4]. The input Brownian trajectories are transformed by the NSDE to time-evolving paths that are then matched to some observed data distribution by means of a discriminator on pathspace. Analogously, NSPDEs can serve as natural generators for spatiotemporal signals; we find of particular interest the definition and computation of an appropriate discrepancy for probability measures supported on spatiotemporal signals, an application we mentioned as future research direction in the paper and that we are currently investigating.
> We would like to emphasise that the NSPDE model is indeed able to integrate partially observed input noise and we refer to the response we gave to reviewer 98nh for a further discussion about this point and additional experiments (also added to section 4.2 of the revised version of the paper, see attached).
>
> Finally, we would like to reiterate (as mentioned in l. 218-224 of the previous version of the paper), that the poor performance on the task $u_0 \mapsto u$ across all models and examples does not constitute a limitation of our model, but rather serves as a sanity check to show that we are not cheating in the experiments by making negligeable the impact of the driving noise on the SPDE solution. Note that in the case of deterministic PDEs (with complete absence of noise) our model performs as well as FNO, as shown in the final example in section B.4 of the appendix.
>
> **Evaluation**
>
> Regarding baselines on neural operators, we used the most recent available implementation of FNO (https://github.com/zongyi-li/fourier_neural_operator), which already underwent significant modifications and improvements compared to the original version.
>
> Concerning the evaluation of FNO, we note that the Hilbert spaces of functions where $u_0$ (and $u$) and the noise $\xi$ are defined could in principle be completely different, therefore it is mathematically difficult to understand how a FNO, as defined in the original paper [5], can consume both objects simultaneously. However, we agree that considering $u_0$ and $\xi$ as generic blocks of data, it is indeed possible to perform a handcrafted modification of these tensors, for example, as you suggested, repeating $u_0$ across the time channel, concatenating this constant path to the path $\xi$ and then applying a FNO to this augmented data stream. We did try to do so, but the performance of the resulting model was very poor across the different experiments.
>
> We ran all experiments with both evaluation methods achieving comparable performance, but we found that the ODE approach is approximately 10 times slower than the Fixed Point approach; this is the main reason why we chose not to include the former in the comparison. We believe that this similarity in performance is to be expected as both methods provide an approximation of the mild solution of the same SPDE in latent space. However, we agree that a thorough comparison between the two evaluation methods would be very beneficial; we included such a discussion in Section 3.5 of the updated version of the paper (see attached).

---

> > ### Author Response · Authors · 2022-08-02
> > **Response to Reviewer QRvu (2/2)**
> >
> > **Writing and presentation**
> >
> > We discuss important motivating examples of SPDEs in the first paragraph of the introduction, explaining how NSPDEs generalise NCDEs to infinite dimensional state spaces, and providing additional details about the considered equations in the experimental section. Nonetheless, we agree that the technical paragraph in l.116-130 (of the previous version of the paper) concerning convergence to a limiting SPDE is perhaps too technical for a machine learning audience, so we propose to move this discussion to the appendix. We also agree that notation in sections 3.1 and 3.2 can be made lighter by transferring some of the details on how to obtain the ODE parameterisation (l. 160-163) to the appendix and by merging equations in l. 180 and 182 into a single equation. We incorporated these changes in the updated version of the paper (see attached)
> >
> > To address the concern about clarity of the method, we added a detailed schematic (Figure 1) to the revised version of the paper (see attached); we hope the latter will facilitate the understanding of how the model operates.
> >
> > We note that it is standard in the SDE/SPDE literature to separate drift and diffusion in the formulation of the equations; for this reason we believe it would be confusing to merge them too early.
> >
> > The related work and novelty of our model are both discussed in the introduction. However, we agree that this might not be very visible without clear headers, which we added to the revised version of the paper (see attached).
> >
> > **References**
> >
> > [1] Crisan, D., Holm, D. D., Luesink, E., Mensah, P. R., & Pan, W. (2021). Theoretical and computational analysis of the thermal quasi-geostrophic model. arXiv preprint arXiv:2106.14850.
> >
> > [2] Cotter, C., Crisan, D., Holm, D. D., Pan, W., & Shevchenko, I. (2019). Numerically modeling stochastic Lie transport in fluid dynamics. Multiscale Modeling & Simulation, 17(1), 192-232.
> >
> > [3] X. Shi, Z. Chen, H. Wang, D.-Y. Yeung, W.-k. Wong, and W.-c. Woo. Convolutional lstm network: A machine learning approach for precipitation nowcasting. In Proceedings of the 28th International Conference on Neural Information Processing Systems - Volume 1, NIPS’15, 802–810. MIT Press, Cambridge, MA, USA, 2015.
> >
> > [4] Kidger, P., Foster, J., Li, X., & Lyons, T. J. (2021, July). Neural sdes as infinite-dimensional gans. In International Conference on Machine Learning (pp. 5453-5463). PMLR.
> >
> > [5] Li, Z., Kovachki, N., Azizzadenesheli, K., Liu, B., Bhattacharya, K., Stuart, A., & Anandkumar, A. (2020). Fourier neural operator for parametric partial differential equations. arXiv preprint arXiv:2010.08895.

---

> ### Comment · Reviewer_QRvu · 2022-08-05
> **Response to Authors - Alleviated Concerns**
>
> I would like to thank the authors for their detailed answer. I think that most of my concerns have been alleviated in the rebuttal:
>  - I still have doubts on the significance of the approach and its applicability outside of the considered experimental setting but found the additional explanations, perspectives and experiments encouraging in this regard.
>  - I still wonder whether more recent baselines should be considered, but my other concerns on the experimental setting have been addressed.
>  - Writing and presentation, although hindered by heavy notations -- which may be unavoidable given this highly technical topic -- have largely been improved, especially thanks to the simplifications and additional diagram.
>
> As a result, I lean towards accepting this paper and consequently raise my score to a 6 - weak accept (but keeping my low confidence given that this is not my primary area of research).

---

### Official Review · Reviewer_YFMX · 2022-07-12

**Rating:** 5
**Confidence:** 1
**Soundness:** 3 good
**Presentation:** 3 good
**Contribution:** 3 good

**Summary:**

The paper introduces the neural stochastic PDE model to learn solution operators of PDEs, extending previous frameworks of neural CDEs and neural operators. The authors introduce two ways of evaluating the model, based on either solving a system of ODEs, or reformulating to a fixed point problem. The authors perform experiments on multiple stochastic PDE problems, showing that their neural stochastic PDE model is superior over previous baseline models in learning such dynamics.

**Questions:**

N/A

**Strengths And Weaknesses:**

Unfortunately, the paper is well outside of my expertise (due to a potential mismatch in the system), and I really do not have the relevant PDE background to analyze the techniques used in the paper.

Aside from the technical parts, I find the structure of the paper pretty clear - the authors provided good motivations for their method, and derived the theory for evaluating the model in an organized fashion. It would be nicer if the authors can summarize their final algorithm in an algorithm block, which would make it much easier for readers to understand the key steps.

---

> ### Author Response · Authors · 2022-08-02
> **Response to Reviewer YFMX**
>
> Thank you for your feedback. To improve the clarity of the method, following your suggestion, we added a detailed schematic (Figure 1) to the revised version of the paper (see attached) describing the key steps of the final algorithm. We hope the latter will facilitate the understanding of how the Neural SPDE model operates.

---

### Meta-Review · Area_Chair_Zcgf · 2022-08-27

**Recommendation:** Accept
**Confidence:** Certain

**Metareview:**

This paper provides an extension of Neural PDEs to Stochastic PDE, which is a very important area lying in the intersection between dynamic systems and stochastic analysis. We think the work is well-motivated, novel, and relevant. The author did a good job in the rebuttal to allieviate most of reviewers' concern about the contribution statement, the evaluation, the presentation, and the adaptivity to irregular sampling. We expect the authors revise the paper accordingly in the next version. Please also explain why there are only two baselines in the paper. It would be great to see some discussions about whether the work can shed some lights on learning the unknown SPDEs, beyond solving them.

**Award:**

No

---

### Decision · Program_Chairs · 2022-09-14

Accept